# NEURAL DIFFUSION MODELS

## ABSTRACT

Diffusion models have shown remarkable performance on many generative tasks. Despite recent success, most diffusion models are restricted in that they only allow linear transformation of the data distribution. In contrast, broader family of transformations can potentially help train generative distributions more efficiently, simplifying the reverse process and closing the gap between the true negative log-likelihood and the variational approximation. In this paper, we present Neural Diffusion Models (NDMs), a generalization of conventional diffusion models that enables defining and learning time-dependent non-linear transformations of data. We show how to optimise NDMs using a variational bound in a simulation-free setting. Moreover, we derive a time-continuous formulation of NDMs, which allows fast and reliable inference using off-the-shelf numerical ODE and SDE solvers. Finally, we demonstrate the utility of NDMs with learnable transformations through experiments on standard image generation benchmarks, including CIFAR-10, downsampled versions of ImageNet and CelebA-HQ. NDMs outperform conventional diffusion models in terms of likelihood and produce high-quality samples.

## 1 INTRODUCTION

Generative models are a powerful class of probabilistic machine learning models with a wide range of applications from e.g. art and music to medicine and physics (Tomczak, 2022; Creswell et al., 2018; Papamakarios et al., 2021; Yang et al., 2022). Generative models learn to mimic the underlying probability distribution of a given data set and can generate novel samples that are similar to the original data. They can for example be used for data augmentation, generating synthetic data sets that increase diversity and scale of the training data, as well as for unsupervised learning, discovering patterns and latent structures in data.

Diffusion models have emerged as a family of generative models that excel at several generative tasks (Sohl-Dickstein et al., 2015; Ho et al., 2020). They parameterize the data model through an iterative refinement process, the *reverse process*, that builds up the data step-by-step from pure noise. For training purposes an auxiliary noising process, the *forward process*, is introduced that successively adds noise to data. The reverse process is then optimized to resemble the forward process. Despite success in various domains (Sohl-Dickstein et al., 2015; Ho et al., 2020; Saharia et al., 2021; Popov et al., 2021; Watson et al., 2022; Trippe et al., 2023), a key limitation of most existing diffusion models is that they rely on a fixed and pre-specified forward process that is unable to adapt to the specific task or data at hand. At the same time there are many works (Hoogeboom & Salimans, 2022; Rombach et al., 2022; Lipman et al., 2022) that improve performance of diffusion models by modifications of the forward processes.

In this paper we develop Neural Diffusion Models (NDMs), a general framework that enables non-linear, time-dependent and learnable data transformations. We extend the approach by Song et al. (2020a) and construct the general forward process as a non-Markovian sequence of latent variables; each latent variable is constructed through a transformation of the data to which we then inject noise. This is then leveraged in the corresponding reverse process. To train NDMs efficiently we generalize the diffusion objective while keeping it a simulation-free bound on the log-likelihood. Furthermore, we derive the time-continuous analogue of the objective function as well as the stochastic differential equation (SDE) and ordinary differential equation (ODE) corresponding to the reverse process.

We demonstrate how NDMs generalizes several existing diffusion models and then propose a new model with learnable transformations of data parameterized by a neural network. To illustrate the

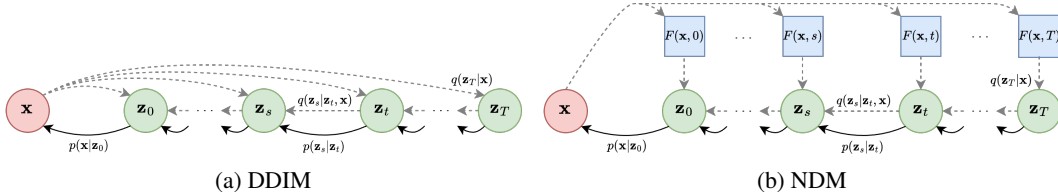

Figure 1: The directed graphical models of DDIM and NDM.

empirical properties of NDMs we provide experimental results on a synthetic data as well as on MNIST, CIFAR-10, downsampled ImageNet and CelebA-HQ image datasets. NDMs consistently outperforms baselines in terms of negative log-likelihood, reaching values of 3.55 and 3.35 on ImageNet 32 and 64 respectively. Moreover, for small to medium number of steps NDMs achieves better image generation quality than denoising diffusion probabilistic models (DDPMs) (Ho et al., 2020), while being comparable for a large number of steps. Finally, we demonstrate that unlike conventional diffusion models, NDMs allows learning simpler generative dynamics like dynamical optimal transport.

We summarize the contributions as follows:

1. We propose neural diffusion models or NDMs, a new framework that generalizes conventional diffusion models in both discrete and continuous time settings.
2. We develop an objective function to optimize NDMs that upper bounds the negative log-likelihood and study its properties.
3. We demonstrate the utility of NDMs with learnable transformations in terms of consistently improved log-likelihood and for small to medium number of steps improved generation quality.

## 2 BACKGROUND

Diffusion models are generative models that make use of latent variables. Given a sample from the data distribution $\mathbf{x} \sim q(\mathbf{x})$, we define a forward noising process that produces latent variables $\mathbf{z}_0, \mathbf{z}_1, \ldots, \mathbf{z}_T$. In contrast, the reverse generative process reverts the forward process, starting by first generating the same latent variables and then data $\mathbf{x}$.

The standard approach to specify the forward process is as a linear Gaussian Markov chain (Sohl-Dickstein et al., 2015; Ho et al., 2020). However, we can also use an implicit definition of the forward process from Song et al. (2020a). This will turn out to be useful for our purposes and is what we focus on here. To construct the implicit forward process we first define the marginal distributions $q(\mathbf{z}_t|\mathbf{x})$. Using these marginal distributions we can define the joint distribution of all latent variables $\mathbf{z}_0, \mathbf{z}_1, \ldots, \mathbf{z}_T$ as follows:

$$q(\mathbf{z}_{0:T}|\mathbf{x}) = q(\mathbf{z}_T|\mathbf{x}) \prod_{t=1}^{T} q(\mathbf{z}_{t-1}|\mathbf{z}_t, \mathbf{x}),$$

$$\text{with} \quad q(\mathbf{z}_{t-1}|\mathbf{z}_t, \mathbf{x}) \quad \text{such that} \quad q(\mathbf{z}_{t-1}|\mathbf{x}) = \int q(\mathbf{z}_t|\mathbf{x}) q(\mathbf{z}_{t-1}|\mathbf{z}_t, \mathbf{x}) d\mathbf{z}_t. \quad (1)$$

Here we make use of the posterior distribution $q(\mathbf{z}_{t-1}|\mathbf{z}_t, \mathbf{x})$ instead of the regular forward distribution $q(\mathbf{z}_t|\mathbf{z}_{t-1})$. Due to the dependence also on the data $\mathbf{x}$ it is a non-Markovian forward process (see Figure 1a). In general the forward process is considered fixed and has no trainable parameters. Moreover, it is specified in such a way that $q(\mathbf{z}_0|\mathbf{x}) \approx \delta(\mathbf{z}_0 - \mathbf{x})$ and $q(\mathbf{z}_T|\mathbf{x}) \approx \mathcal{N}(\mathbf{z}_T; 0, I)$. So if $q(\mathbf{z}_{t-1}|\mathbf{z}_t)$ was available we could sample $\mathbf{z}_T \sim \mathcal{N}(\mathbf{z}_T; 0, I)$ and run the reverse process to get $\mathbf{z}_0 \sim q(\mathbf{z}_0) \approx q(\mathbf{x})$. However, the distribution $q(\mathbf{z}_{t-1}|\mathbf{z}_t)$ depends implicitly on the data distribution $q(\mathbf{x})$ and thus has a complicated form, so we instead approximate the reverse process using a Markov chain with distribution $p_\theta(\mathbf{z}_{0:T})$:

$$p_\theta(\mathbf{z}_{0:T}) = p(\mathbf{z}_T) \prod_{t=1}^{T} p_\theta(\mathbf{z}_{t-1}|\mathbf{z}_t), \quad \text{where} \quad p(\mathbf{z}_T) = \mathcal{N}(\mathbf{z}_T; 0, I). \quad (2)$$

The combination of the forward process $q$ and the reverse process $p_\theta$ is a form of (hierarchical) variational autoencoder (Kingma & Welling, 2013; Rezende et al., 2014). Therefore, it can be trained by optimizing the usual variational bound on the negative log-likelihood. In the case of diffusion models, it can be written as follows (see Section A of Ho et al. (2020)):

$$\mathbb{E}_q \left[ \underbrace{D_{\mathrm{KL}}\Big(q(\mathbf{z}_T|\mathbf{x})||p(\mathbf{z}_T)\Big)}_{\mathcal{L}_{\mathrm{prior}}} + \underbrace{\sum_{t=1}^{T} D_{\mathrm{KL}}\Big(q(\mathbf{z}_{t-1}|\mathbf{z}_t,\mathbf{x})||p_\theta(\mathbf{z}_{t-1}|\mathbf{z}_t)\Big)}_{\mathcal{L}_{\mathrm{diff}}} - \underbrace{\log p_\theta(\mathbf{x}|\mathbf{z}_0)}_{\mathcal{L}_{\mathrm{rec}}} \right]. \quad (3)$$

Since the process $q$ and the distribution $p_\theta(\mathbf{z}_T) = p(\mathbf{z}_T)$ are fixed, the prior term $\mathcal{L}_{\mathrm{prior}}$ does not depend on the parameters $\theta$, so it can be omitted. The distribution $p_\theta(\mathbf{x}|\mathbf{z}_0)$ is often take to be a Gaussian distribution, with low variance, for continuous data and a dequantization distribution for discrete data. Thus, also the reconstruction term $\mathcal{L}_{\mathrm{rec}}$ does not depend on the parameter $\theta$.

This means that the only part that depends on the model parameters $\theta$ is the diffusion term $\mathcal{L}_{\mathrm{diff}}$. It is a sum of Kullback–Leibler (KL) divergences between posterior distributions in the forward process $q(\mathbf{z}_{t-1}|\mathbf{z}_t,\mathbf{x})$ and the distributions $p_\theta(\mathbf{z}_{t-1}|\mathbf{z}_t)$ from the reverse process. In the general case this KL divergence is intractable, so the standard choice here is to set the marginal conditional distributions to be Gaussian, i.e. $q(\mathbf{z}_t|\mathbf{x}) = \mathcal{N}(\mathbf{z}_t; \alpha_t\mathbf{x}, \sigma_t^2 I)$. The posterior distribution then takes the form:

$$q(\mathbf{z}_s|\mathbf{z}_t,\mathbf{x}) = \mathcal{N}\left(\mathbf{z}_s; \alpha_s\mathbf{x} + \frac{\sqrt{\sigma_s^2 - \tilde{\sigma}_{s|t}^2}}{\sigma_t}(\mathbf{z}_t - \alpha_t\mathbf{x}), \tilde{\sigma}_{s|t}^2 I\right), \quad \text{for} \quad 0 \le s \le t \le T. \quad (4)$$

Note that here we allow for an arbitrary choice of time grid, i.e. $s$ and $t$, whereas above it was equidistant. It is straightforward to check that such a posterior distribution satisfies (1) for any $\tilde{\sigma}_{s|t}^2 \le \sigma_s^2$. The exact schedule of $\tilde{\sigma}_{s|t}^2$ is a user design choice.

Finally, the reverse distribution is set to $p_\theta(\mathbf{z}_s|\mathbf{z}_t) = q(\mathbf{z}_s|\mathbf{z}_t, \hat{\mathbf{x}}_\theta(\mathbf{z}_t, t))$, where $\hat{\mathbf{x}}_\theta(\mathbf{z}_t, t)$ is the model's prediction of $\mathbf{x}$. Since $q(\mathbf{z}_s|\mathbf{z}_t,\mathbf{x})$ and $p_\theta(\mathbf{z}_s|\mathbf{z}_t)$ are both Gaussian distributions, we can compute the KL divergences in $\mathcal{L}_{\mathrm{diff}}$ in closed form.

This choice of forward and reverse processes, resulting in analytic expressions for the diffusion terms given data, is what makes diffusion models a *simulation-free* approach. Simulation-free means that we do not have to sample all latent variables for each optimization step. Rather than calculating all individual terms in $\mathcal{L}_{\mathrm{diff}}$, we can uniformly sample $t$ and optimize only a subset of KL divergences using stochastic gradient descent.

By choosing a specific value for $\tilde{\sigma}_{s|t}^2$, we can obtain equality between the processes of DDPM and DDIM (see section 4.1 of Song et al. (2020a)). Furthermore, as Song et al. (2020b) demonstrated, when the number of steps $T$ in DDPM goes to infinity, we can transition to continuous time. In this scenario, the reverse process can be described using a Stochastic Differential Equation (SDE):

$$d\mathbf{z}_t = [r(t)\mathbf{z}_t - g^2(t)s_\theta(\mathbf{z}_t,t)]dt + g(t)d\mathbf{w}_t, \quad \text{where} \quad s_\theta(\mathbf{z}_t,t) = \frac{\alpha_t\hat{\mathbf{x}}_\theta(\mathbf{z}_t,t) - \mathbf{z}_t}{\sigma_t^2} \quad (5)$$

$$r(t) = \frac{d\log\alpha_t}{dt} \quad \text{and} \quad g^2(t) = \frac{d\sigma_t^2}{dt} - 2\frac{d\log\alpha_t}{dt}\sigma_t^2, \quad (6)$$

with time running backwards from $t = 1$ to $t = 0$. This formulation allows us to switch to the equivalent ODE and to use different SDE and ODE solvers for sampling and density estimation.

## 3   NEURAL DIFFUSION MODELS

Diffusion models can be viewed as a special type of hierarchical variational autoencoders, where the latent variables are inferred using scaling of data points and injecting of Gaussian noise. However, this formulation limits diffusion models in terms of the flexibility of the latent space, which prevents from learning more useful distributions for the reverse (generative) process. To overcome this limitation, we propose a general form of transformations of data that allows to define and learn distributions of the latent space.

Table 1: Summary of existing diffusion models as instances of Neural Diffusion Models (NDM). See extended table in Appendix B.

| Model | Distribution $q(\mathbf{z}_t|x)$ | NDM's $F(\mathbf{x}, t)$ | Comment |
|---|---|---|---|
| DDPM (Ho et al., 2020) / DDIM (Song et al., 2020a) | $\mathcal{N}\left(\mathbf{z}_t; \alpha_t\mathbf{x}, \sigma_t^2 I\right)$ | $\mathbf{x}$ | |
| Flow Matching OT (Lipman et al., 2022) | $\mathcal{N}\left(\mathbf{z}_t; \alpha_t\mathbf{x}, \sigma_t^2 I\right)$ | $\mathbf{x}$ | $\alpha_t = t$, $\sigma_t = 1 - (1 - \sigma_{\min})t$ |
| VDM (Kingma et al., 2021) | $\mathcal{N}\left(\mathbf{z}_t; \alpha_t\mathbf{x}, \sigma_t^2 I\right)$ | $\mathbf{x}$ | $\alpha_t^2 = \mathrm{sigmoid}(-\gamma_\eta(t))$, $\sigma_t^2 = \mathrm{sigmoid}(\gamma_\eta(t))$ |
| Soft Diffusion (Daras et al., 2022) | $\mathcal{N}\left(\mathbf{z}_t; C_t\mathbf{x}, s_t^2 I\right)$ | $C_t\mathbf{x}$ | $\alpha_t = 1, \sigma_t^2 = s_t^2$ |
| LSGM (Vahdat et al., 2021) | $\mathcal{N}\left(\mathbf{z}_t; \alpha_t E(\mathbf{x}), \sigma_t^2 I\right)$ | $E(\mathbf{x})$ | $p(x|z_0) = \mathcal{N}\left(x; aD(z_0), \sigma^2\right)$ |

In this section, we introduce the Neural Diffusion Models (NDMs) – a simulation-free framework that generalises conventional diffusion models. The key idea in NDMs is to apply a time-dependent transformation $F_\varphi(\mathbf{x}, t)$ to the data $\mathbf{x}$ at each step of forward process before injecting noise. Previous diffusion models arise as special cases when the data transformation is either linear, time-independent, or pre-defined non-linear (see Table 1). In contrast, the NDM can work with any time-dependent transformation of data and may be learned end-to-end. In Section 4 we provide experimental results with $F_\varphi(\mathbf{x}, t)$ parameterized by neural networks.

### 3.1 MODEL DEFINITION AND VARIATIONAL OBJECTIVE

We introduce NDMs constructively. First, we define the desired marginal distributions:

$$q_\varphi(\mathbf{z}_t|\mathbf{x}) = \mathcal{N}\left(\mathbf{z}_t; \alpha_t F_\varphi(\mathbf{x}, t), \sigma_t^2 I\right), \tag{7}$$

where $F_\varphi(\mathbf{x}, t) : \mathbb{R}^d \times [0, T] \mapsto \mathbb{R}^d$ is a function parameterized by $\varphi$ that applies a time-dependent transform to the data point $\mathbf{x}$. We adapt the approach from DDIM, as described in Section 2, and choose the following posterior distribution that satisfies (7) (we provide derivation and proof in Appendix A.1):

$$q_\varphi(\mathbf{z}_s|\mathbf{z}_t, \mathbf{x}) = \mathcal{N}\left(\mathbf{z}_s; \alpha_s F_\varphi(\mathbf{x}, s) + \frac{\sqrt{\sigma_s^2 - \tilde{\sigma}_{s|t}^2}}{\sigma_t}\Big(\mathbf{z}_t - \alpha_t F_\varphi(\mathbf{x}, t)\Big), \tilde{\sigma}_{s|t}^2 I\right), \tag{8}$$

for $0 \leq s \leq t \leq T$ where $\tilde{\sigma}_{s|t}^2 \leq \sigma_s^2$ is a design choice. Using this posterior we can define an implicit forward process according to (1) (see Figure 1b). This forward process provides access to both marginal and posterior distributions just like in the DDIM framework (Song et al., 2020a). The corresponding NDM variational objective has the following form:

$$\mathbb{E}_{q_\varphi}\left[\underbrace{\mathrm{D}_{\mathrm{KL}}\Big(q_\varphi(\mathbf{z}_T|\mathbf{x})||p(\mathbf{z}_T)\Big)}_{\mathcal{L}_{\mathrm{prior}}} + \underbrace{\sum_{t=1}^{T} \mathrm{D}_{\mathrm{KL}}\Big(q_\varphi(\mathbf{z}_{t-1}|\mathbf{z}_t, \mathbf{x})||p_\theta(\mathbf{z}_{t-1}|\mathbf{z}_t)\Big)}_{\mathcal{L}_{\mathrm{diff}}} - \underbrace{\log p_\theta(\mathbf{x}|\mathbf{z}_0)}_{\mathcal{L}_{\mathrm{rec}}}\right]. \tag{9}$$

While the objective has the same form as in DDIM (3), the individual terms are different. If the transformation $F_\varphi(\mathbf{x}, t)$ is actually parameterized by learnable parameters $\varphi$, the prior term $\mathcal{L}_{\mathrm{prior}}$ and the reconstruction term $\mathcal{L}_{\mathrm{rec}}$ depend on the parameter $\varphi$ as well. Therefore, in that case these terms cannot be excluded from the optimization process.

---

**Algorithm 1** Learning NDM

**Require:** $q(\mathbf{x})$, $F_\varphi$, $\hat{\mathbf{x}}_\theta$
  **for** learning iterations **do**
    $\mathbf{x} \sim q(\mathbf{x})$, $t \sim U[1, T]$, $\varepsilon \sim \mathcal{N}(0, I)$
    $\mathbf{z}_t \sim q_\varphi(\mathbf{z}_t|\mathbf{x})$
    $\mathcal{L} = \mathcal{L}_{\text{rec}} + \mathcal{L}_{\text{diff}} + \mathcal{L}_{\text{prior}}$
    Gradient step on $\theta$ and $\varphi$ w.r.t. $\mathcal{L}$
  **end for**

---

**Algorithm 2** Sampling from NDM

**Require:** $F_\varphi$, $\hat{\mathbf{x}}_\theta$
  $\mathbf{z}_T \sim \mathcal{N}(0, I)$
  **for** $t = T, \ldots, 1$ **do**
    $\hat{\mathbf{x}} = \hat{\mathbf{x}}_\theta(\mathbf{z}_t, t)$
    $\mathbf{z}_{t-1} \sim q_\varphi(\mathbf{z}_{t-1}|\mathbf{z}_t, \hat{\mathbf{x}})$
  **end for**
  **return** $\mathbf{z}_0$

---

For the standard parameterization of the reverse process through approximate posteriors $p_\theta(\mathbf{z}_s|\mathbf{z}_t) = q_\varphi(\mathbf{z}_s|\mathbf{z}_t, \hat{\mathbf{x}}_\theta(\mathbf{z}_t, t))$ the KL divergences in the diffusion term $\mathcal{L}_{\text{diff}}$ are (see Appendix A.2):

$$\mathrm{D}_{\mathrm{KL}}\Big(q_\varphi(\mathbf{z}_s|\mathbf{z}_t, \mathbf{x})||p_\theta(\mathbf{z}_s|\mathbf{z}_t)\Big) =$$

$$\frac{1}{2\tilde{\sigma}_{s|t}^2}\left\|\alpha_s\Big(F_\varphi(\mathbf{x}, s) - F_\varphi(\hat{\mathbf{x}}_\theta(\mathbf{z}_t, t), s)\Big) + \frac{\sqrt{\sigma_s^2 - \tilde{\sigma}_{s|t}^2}}{\sigma_t}\alpha_t\Big(F_\varphi(\hat{\mathbf{x}}_\theta(\mathbf{z}_t, t), t) - F_\varphi(\mathbf{x}, t)\Big)\right\|_2^2. \quad (10)$$

Note a distinction between the objectives of NDM and DDIM here. In the case of DDIM, the model tries to accurately predict the data point $\mathbf{x}$. In contrast, NDM aims to predict the *transformed* data point $F_\varphi(\mathbf{x}, t)$. Despite this change, NDM's optimization remains simulation-free, so we can efficiently train the NDM by sampling time steps and calculating corresponding KL divergences. We summarise the training and sampling procedures in Algorithms 1 and 2.

Given that NDM is a generalization of DDIM, we can leverage the same techniques for inference. Specifically, we can adjust the number of intermediate time steps, the schedule of $\tilde{\sigma}_{s|t}^2$ as well as sampling with various dynamics, including a deterministic dynamic corresponding to $\tilde{\sigma}_{s|t}^2 = 0$.

## 3.2 CONTINUOUS TIME NDMS

We previously formulated NDMs in the discrete time setting with $T$ steps. However, like conventional diffusion models, we can let the number of steps $T$ go to infinity and switch to continuous time. In this case, the set of time steps $\{0, 1, \ldots, T\}$ transforms to the range $[0, 1]$ and the diffusion term of the objective reduces to an expectation over time (see derivation in Appendix A.4):

$$\mathcal{L}_{\text{diff}} = \mathbb{E}_{q(\mathbf{x})}\mathbb{E}_{u(t)}\mathbb{E}_{q(\mathbf{z}_t|\mathbf{x})}\frac{1}{g^2(t)}\left\|\alpha_t\Big(\dot{F}_\varphi(\mathbf{x}, t) - \dot{F}_\varphi\big(\hat{\mathbf{x}}_\theta(\mathbf{z}_t, t), t\big)\Big) + \right.$$

$$\left. \frac{1}{2}\left(\frac{\partial \sigma_t^2}{\partial t} - 2r(t)\sigma_t^2 + g^2(t)\right)\Big(s(\mathbf{x}, \mathbf{z}_t, t) - s\big(\hat{\mathbf{x}}_\theta(\mathbf{z}_t, t), \mathbf{z}_t, t\big)\Big)\right\|_2^2,$$

$$\text{where} \quad r(t) = \frac{\partial \log \alpha_t}{\partial t}, \quad g^2(t) = \dot{\nu}_t\sigma_t^2 \quad \text{and} \quad s(\mathbf{x}, \mathbf{z}_t, t) = \frac{\alpha_t F_\varphi(\mathbf{x}, t) - \mathbf{z}_t}{\sigma_t^2}. \quad (11)$$

Similar to training a discrete time NDM, we can train a continuous time NDMs by sampling time. In our experiments we use importance sampling (Song et al., 2021) and sample time from a distribution proportional to $\frac{1}{g^2(t)}$.

Note, that we may not have access to the partial derivative of the transformation $F_\varphi(\cdot, t)$ with respect to $t$ in closed form. However, for any differentiable $F_\varphi(\cdot, t)$ we can use Jacobian-Vector product (Hirsch et al., 2012) to obtain this derivative.

The discrete time reverse process also becomes a continuous time process, described by a Stochastic Differential Equation (SDE). If we parameterize the noise injection in the posterior distribution as

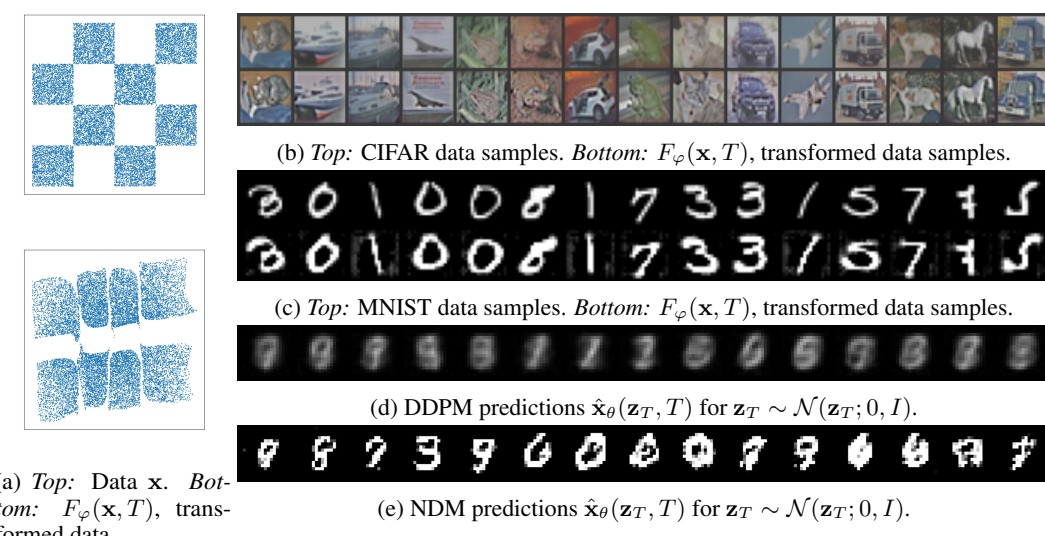

(b) *Top:* CIFAR data samples. *Bottom:* $F_\varphi(\mathbf{x}, T)$, transformed data samples.

(c) *Top:* MNIST data samples. *Bottom:* $F_\varphi(\mathbf{x}, T)$, transformed data samples.

(d) DDPM predictions $\hat{\mathbf{x}}_\theta(\mathbf{z}_T, T)$ for $\mathbf{z}_T \sim \mathcal{N}(\mathbf{z}_T; 0, I)$.

(a) *Top:* Data $\mathbf{x}$. *Bottom:* $F_\varphi(\mathbf{x}, T)$, transformed data.

(e) NDM predictions $\hat{\mathbf{x}}_\theta(\mathbf{z}_T, T)$ for $\mathbf{z}_T \sim \mathcal{N}(\mathbf{z}_T; 0, I)$.

Figure 2: Learned transforms for the 2D checkerboard distribution (*left*). Learned transforms for CIFAR-10 and MNIST (*top right*), as well as predictions for MNIST (*bottom right*). NDM learns useful forward transformations and more accurately predicts the data from injected noise.

$\tilde{\sigma}_{s|t}^2 = \sigma_s^2(1 - e^{\nu_s - \nu_t})$, we obtain the following SDE (see derivation in Appendix A.3):

$$d\mathbf{z}_t = \left[\alpha_t \dot{F}_\varphi(\hat{\mathbf{x}}_\theta(\mathbf{z}_t, t), t) + r(t)\mathbf{z}_t - \frac{1}{2}\Big(g^2(t) - 2r(t)\sigma_t^2\Big)s_\theta(\mathbf{z}_t, t)\right] dt + g(t)d\mathbf{w}, \quad (12)$$

$$\text{where} \quad s_\theta(\mathbf{z}_t, t) = \frac{\alpha_t F_\varphi(\hat{\mathbf{x}}_\theta(\mathbf{z}_t, t), t) - \mathbf{z}_t}{\sigma_t^2}. \quad (13)$$

By changing the function $\nu_t$, we can obtain different dynamics. In the extreme case where $\nu_t$ is equal to a constant we have deterministic dynamics described by an ODE. This enables the use of SDE or ODE solvers for inference. Moreover, we can estimate densities by considering the model as a continuous normalizing flow (Chen et al., 2018) in the deterministic case.

## 4 EXPERIMENTS

We present empirical results for the proposed Neural Diffusion Models with learnable transformations on a synthetic datasets as well as multiple image datasets. Qualitatively, NDMs learn transformations that simplify the data distribution, leading to predictions of $\mathbf{x}$ that are more aligned with the data. Quantitatively, NDMs consistently outperform the baseline in terms of likelihood. Moreover, for a small to medium number of steps, NDMs achieve better image generation quality than DDPM, while being comparable for a large number of steps. Additionally we provide a proof of concept experiment that demonstrates the ability of NDMs to learn simple generative trajectories.

### 4.1 IMPLEMENTATION DETAILS

We demonstrate NDMs with learnable transformations on the MNIST (Deng, 2012), CIFAR-10 (Krizhevsky et al., 2009), downsampled ImageNet (Deng et al., 2009; Chrabaszcz et al., 2017) and CelebA-HQ-256 (Karras et al., 2017) datasets. In all experiments we use same neural network architectures to parameterize both the generative process and the transformations $F_\varphi$. In experiments with images we use the U-Net architecture from Dhariwal & Nichol (2021). To ensure consistency with Song et al. (2020b; 2021), we apply horizontal flipping as a data augmentation technique for training models on CIFAR-10 and ImageNet. Unless otherwise stated, we utilize the DDPM variance-preserving schedule of noise injection for $\alpha_t$ and $\sigma_t^2$. For density estimation of discrete data we use uniform dequantization.

Table 2: Summary of our findings for density modeling tasks.

| Model | CIFAR10 | ImageNet 32 | ImageNet 64 |
|---|---|---|---|
| DDPM (Ho et al., 2020) | 3.69 | | |
| Improved DDPM (Nichol & Dhariwal, 2021) | 2.94 | | 3.54 |
| VDM (Kingma et al., 2021) | **2.65** | 3.72 | 3.40 |
| Score SDE (Song et al., 2020b) | 2.99 | | |
| Score Flow (Song et al., 2021) | 2.83 | 3.76 | |
| NDM (ours) | 2.70 | **3.55** | **3.35** |

Table 3: Generative results on CelebA-HQ-256 for LSGM and NDM with learnable transformations in the latent space of VAE.

| Model | NLL ↓ | FID ↓ |
|---|---|---|
| LSGM (Vahdat et al., 2021) | $\leq 0.70$ | 7.22 |
| Latent NDM (ours) | $\leq \mathbf{0.65}$ | **7.18** |

In the experiments we report negative log-likelihood (NLL) in bits per dimension (BPD), negative evidence lower bound (NELBO) (9), and sample quality as measured by the Frechet Inception Distance (FID) (Heusel et al., 2017). We calculate NLL by integrating the corresponding ODEs using the RK45 solver from Dormand & Prince (1980), and both NLL and NELBO are calculated on test data. For FID we report the score averaged over 50k images.

In Section 3 we parameterize the reverse process through $\hat{x}_\theta$ function. However, in practice we reparameterize the generative process in terms of prediction of injected noise. For a detailed description of parameterizations and other experimental details, please refer to Appendix C.

### 4.2 LEARNED TRANSFORMATIONS

Let us examine some of the transformations that NDM learns. Figure 2a-2c illustrates the transformations that NDM learns for the 2D checkerboard distribution, MNIST, and CIFAR-10 datasets. For the checkerboard, we observe that $F_\varphi$ learns to transform the interleaved pattern into a non-interleaved one. In the case of the grayscale digits of the MNIST dataset, $F_\varphi$ learns to highlight the distinctive features of the numbers. It thickens the lines and even creates bubbles at the corners. For the color images of CIFAR-10, $F_\varphi$ learns to increase the image contrast. In all cases, our model learns a way to simplify the data distribution. These transformations may enable the reverse process to transition more smoothly from simple distributions to complex ones.

Furthermore, we would like to emphasize the difference between the predictions of $x$ that NDM and DDPM makes. Figure 2d and Figure 2e shows the predictions $\hat{x}_\theta(z_T, T)$ generated by NDM and DDPM models trained on the MNIST dataset. In each case, the model samples from a standard normal distribution $z_T \sim \mathcal{N}(z_T; 0, I)$ and based on this value tries to predict $x$. Therefore, we do not expect these samples to be of high quality. However, as we can see, NDM's predictions of $x$ are much more similar to the data distribution than DDPM's predictions. We attribute this behavior to the fact that our model aims to predict not the datapoint $x$, but the transformed datapoint $F_\varphi(x, t)$. Thus, to make better predictions of the transformed datapoint, it may be critical to generate predictions of $x$ that resemble real data. Any deviation from the $x$-distribution is exaggerated by the transformation and thus less likely to happen for NDM's predictions.

In Appendix D we provide additional samples for terminal and intermediate timestaps.

### 4.3 IMAGE GENERATION

Next, we evaluate NDMs with learnable transformations quantitatively. We train continuous time NDM on MNIST, CIFAR-10, and downsampled ImageNet datasets. Table 2 summarizes our results, reporting NLL. NDMs demonstrates performance on CIFAR-10 that is comparable with the baselines and outperforms baselines on ImageNet.

Table 4: Performance comparison of the DDPM and NDM on CIFAR-10 and ImageNet 32 datasets. We report FID scores for DDPM-style (FID) and DDIM-style (FID*) sampling procedures.

| Steps | Model | CIFAR-10 | | | | ImageNet 32 | | | |
|---|---|---|---|---|---|---|---|---|---|
| | | NLL ↓ | NELBO ↓ | FID ↓ | FID* ↓ | NLL ↓ | NELBO ↓ | FID ↓ | FID* ↓ |
| 1000 | DDPM | 3.11 | 3.18 | **11.44** | **13.35** | 3.89 | 3.95 | **16.18** | **19.08** |
| | NDM | **3.02** | **3.03** | 11.82 | 13.79 | **3.79** | **3.82** | 17.02 | 19.76 |
| 10 | DDPM | 5.02 | 5.13 | 37.83 | **19.89** | 6.28 | 6.42 | 53.51 | **26.47** |
| | NDM | **4.63** | **4.74** | **31.56** | 22.20 | **5.81** | **5.94** | **45.38** | 29.95 |
| 1000 → 10 | DDPM | 8.78 | 8.98 | **43.85** | 17.73 | 10.99 | 11.23 | **58.35** | 25.53 |
| | NDM | **8.58** | **8.81** | 48.41 | **16.96** | **10.78** | **11.06** | 62.12 | **23.77** |

Then, we compare NDM with the DDPM baseline on MNIST, CIFAR-10, and ImageNet 32 datasets. To ensure a fair comparison, when implementing DDPM we use an NDM with fixed identity transformation $F_\varphi(\mathbf{x}, t) = \mathbf{x}$. Therefore, we train both models with the same objective (9) and hyperparameters. The first part of Table 4 summarizes our results, reporting NLL, NELBO (9), and FID score. NDM demonstrates comparable sample quality with the baseline on all datasets and consistently outperforms the baseline on NLL and NELBO, especially for smaller numbers of steps. This improvement may be attributed to NDM's ability to fit distributions of the forward process and simplify the denoising task for the reverse process.

We also compare NDM with DDPM in a setup where we train both models with $T = 1000$ steps and then sample with fewer steps. The second part of Table 4 summarizes our results, which are consistent with the corresponding numbers of steps used during training. However, in absolute values, both models show worse performance when we decrease the number of steps, and NDM demonstrates a more severe degradation. This observation is especially noticeable for small numbers of steps, such as $T = 10$, where NDM has a better FID score than DDPM when trained with 10 steps, but a worse FID score when the number of steps is decreased from 1000 to 10. From this, we conclude that although NDM can in principle work with reduced number of steps it is less robust to such modifications compared to DDPM.

Finally, we demonstrate that NDMs may be successfully combined with LSGM (Vahdat et al., 2021). For this experiment we replaced the linear diffusion in the LSGM baseline for CelebA-HQ-256 with NDMs featuring the learnable $F_\varphi$. We parameterise $F_\varphi$ with the same neural network architecture as baseline's architecture for parameterisation of diffusion. Table 3 demonstrates that NDMs have better likelihood estimation and sample quality.

In Appendix B.5 we provide further discussion and in Appendix D we provide additional results and ablation studies.

## 5 RELATED WORK

NDMs build on diffusion probabilistic models originally proposed by Sohl-Dickstein et al. (2015), which can be considered as an instance of (hierarchical) variational autoencoders (VAEs) (Kingma & Welling, 2013; Rezende et al., 2014). Recently, the theory of diffusion models was extended to deterministic sampling (Song et al., 2020a) and continuous time (Song et al., 2020b). These results allowed to reach impressive performance in image generation tasks (Ho et al., 2020; Song et al., 2020b; Dhariwal & Nichol, 2021; Kingma et al., 2021). However, most existing diffusion models have a significant limitation in that they rely on a pre-specified and simple noise injection process that is unable to adapt to the specific task or data at hand. To overcome this, researchers have explored ways to generalize diffusion models.

Various papers have since proposed ways to speed up sampling from diffusion models. Tachibana et al. (2021) and Liu et al. (2022a) proposed alternative SDE and ODE solvers. Xiao et al. (2021) proposed replacing simple Gaussian distributions at each generation step with distributions learned by GANs (Goodfellow et al., 2014). Some works proposed methods like iterative distillation with a reduction in the number of steps (Salimans & Ho, 2022) and iterative straightening of trajectories (Liu et al., 2022b; Liu, 2022). While these methods change the generative process, they are compatible with NDMs.

Several papers proposed constructing the process of data corruption not by noise injection, but rather by blurring (Rissanen et al., 2022; Daras et al., 2022; Hoogeboom & Salimans, 2022) or through another linear transformation (Singhal et al., 2023). Another line of work modifies directly the dynamics of diffusion models through mapping the data into the latent space of VAE (Vahdat et al., 2021; Rombach et al., 2022), hierarchical VAE (Gu et al., 2022) or normalizing flow (Kim et al., 2022) models and then runs standard linear diffusion. As we demonstrate in Tables 1, these arise as distinct special cases of NDMs for specific choices of the transformation $F_\varphi$.

Inspired by diffusion models, several works (Lipman et al., 2022; Neklyudov et al., 2022) have proposed simulation-free objectives for training continuous normalizing flows. These approaches are similar to diffusion models as they rely on the idea of reversing a predefined corruption process. Later, some works (Albergo & Vanden-Eijnden, 2022; Lee et al., 2023) extended these ideas and proposed to learn the forward process. However, although NDMs and these works are similar in spirit, they differ in that they optimize the forward process specifically to obtain straight generative trajectories, while in our approach we optimize learnable forward process to minimize variational bound on NLL, which not necessarily leads to straight generative trajectories.

In another line of works (De Bortoli et al., 2021; Wang et al., 2021; Peluchetti), finite-time diffusion constructions were proposed using diffusion bridge theory to address the approximation error incurred by infinite-time denoising constructions. While such approaches allow learning forward transformations, they require inferring all latent variables for each optimization step. This limitation break the simulation-free paradigm and can make these models expensive to train. NDM in contrast allows learning forward transformations efficiently and simulation-free.

In Appendix B, we provide further discussion and comparisons with related works.

## 6 LIMITATIONS

Compared to conventional diffusion models, NDMs with learnable transformations have twice as many parameters, which results in slower training. Specifically, in experiments on images, NDMs with learnable transformations take approximately 2.3 times longer than DDPM to train. However, no additional techniques where necessary to ensure stable training of NDMs. Additionally, in Appendix D, we provide an ablation study demonstrating that performance improvements are not achieved by increasing the number of parameters.

Another distinction between NDMs and DDPM is the importantance for NDMs in using the full objective (9) when training the model. A simplified objective, such as $\mathcal{L}_{\text{simple}}$ used in DDPM, which measures how well the model predicts injected noise and does not take into account the transformation $F_\varphi$, can cause the collapse of this transformation to 0. The reason for this is that it becomes trivial to identify the injected noise through $\mathbf{z}_t$.

Finally, unlike conventional diffusion, the generative process of NDMs with learnable transformations depends on the parameters of the forward process. Therefore, in the case of learnable parameters, NDMs do not support conditional generation techniques with classifier guidance (Dhariwal & Nichol, 2021). However, we can utilize alternative approaches (Wu et al., 2023) to enable conditional generation from NDMs, but we will defer this to future research.

## 7 CONCLUSION

We introduced Neural Diffusion Models (NDMs), a new class of diffusion models that enables defining and learning the general forward noising process. First, we showed how to optimize NDMs using a variational bound in a simulation-free setting. Then, we derived a time-continuous formulation of NDMs allowing for fast and reliable inference and likelihood evaluation using off-the-shelf numerial ODE and SDE solvers. Next, we demonstrated how some existing diffusion models appear as a special cases of NDMs. For NDMs with learnable transformations we studied their utility on standard image generation benchmarks. NDMs outperforms conventional diffusion models in terms of likelihood and produces samples of comparable quality.

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
