## A DERIVATIONS AND PROOFS

### A.1 FORWARD POSTERIOR

First, we rewrite the marginal distribution (7) in terms of standard normally distributed $\varepsilon_t$, $\varepsilon_s$ for $s$ and $t$, where $s < t$:

$$\mathbf{z}_t = \alpha_t F_\varphi(\mathbf{x}, t) + \sigma_t \varepsilon_t, \tag{14}$$

$$\mathbf{z}_s = \alpha_s F_\varphi(\mathbf{x}, s) + \sigma_s \varepsilon_s. \tag{15}$$

Next, we constructively introduce the posterior distribution $q_\varphi(\mathbf{z}_s|\mathbf{z}_t, \mathbf{x})$. To sample $\mathbf{z}_s$ given $\mathbf{z}_t$ and $\mathbf{x}$ while preserving the correct marginal distribution $q_\varphi(\mathbf{z}_s|\mathbf{x})$, we can combine the noise $\varepsilon_t$ with additional noise $\tilde{\varepsilon}_{s|t}$ as follows:

$$\mathbf{z}_s = \alpha_s F_\varphi(\mathbf{x}, s) + \sqrt{\sigma_s^2 - \tilde{\sigma}_{s|t}^2}\, \varepsilon_t + \tilde{\sigma}_{s|t}\tilde{\varepsilon}_{s|t}. \tag{16}$$

The samples $\mathbf{z}_s$ follow a (conditional) normal distribution. By marginalizing $\varepsilon_t$ and $\tilde{\varepsilon}_{s|t}$, we obtain a normal distribution with mean $\alpha_s F\varphi(\mathbf{x}, s)$ and variance $\sigma_s^2 - \tilde{\sigma}_{s|t}^2 + \tilde{\sigma}_{s|t}^2 = \sigma_s^2$. Therefore, this sampling procedure satisfies $q_\varphi(\mathbf{z}_s|\mathbf{x}) = \int q_\varphi(\mathbf{z}_t|\mathbf{x})q_\varphi(\mathbf{z}_s|\mathbf{z}_t, \mathbf{x})d\mathbf{z}_t$.

The equation (16) relies on $\varepsilon_t$, which we do not have explicit access to. However, once we know $\mathbf{z}_t$ and $\mathbf{x}$, we can calculate it from (14) as $\varepsilon_t = \frac{\mathbf{z}_t - \alpha_t F_\varphi(\mathbf{x},t)}{\sigma_t}$ and substitute it in (16):

$$\mathbf{z}_s = \alpha_s F_\varphi(\mathbf{x}, s) + \frac{\sqrt{\sigma_s^2 - \tilde{\sigma}_{s|t}^2}}{\sigma_t}\Big(\mathbf{z}_t - \alpha_t F_\varphi(\mathbf{x}, t)\Big) + \tilde{\sigma}_{s|t}\tilde{\varepsilon}_{s|t}. \tag{17}$$

Using this constructive definition, we obtain the posterior distribution (8).

### A.2 OBJECTIVE

To calculate the diffusion term $\mathcal{L}_{\text{diff}}$ (10) of the objective, we need to compute the KL divergence between the forward posterior distribution $q_\varphi(\mathbf{z}_s|\mathbf{z}_t, \mathbf{x})$ and the reverse distribution $p_\theta(\mathbf{z}_s|\mathbf{z}_t)$. Since we use parameterization $p_\theta(\mathbf{z}_s|\mathbf{z}_t) = q_\varphi(\mathbf{z}_s|\mathbf{z}_t, \hat{\mathbf{x}}_\theta(\mathbf{z}_t, t))$, both of these distributions are normal distributions with the same variance, so we can evaluate the KL divergence between them analytically as follows:

$$D_{\text{KL}}\Big(q_\varphi(\mathbf{z}_s|\mathbf{z}_t, \mathbf{x})||p_\theta(\mathbf{z}_s|\mathbf{z}_t)\Big) =$$

$$= \frac{1}{2\tilde{\sigma}_{s|t}^2}\Big\|\alpha_s F_\varphi(\mathbf{x}, s) + \frac{\sqrt{\sigma_s^2 - \tilde{\sigma}_{s|t}^2}}{\sigma_t}\Big(\cancel{\mathbf{z}_t} - \alpha_t F_\varphi(\mathbf{x}, t)\Big) -$$

$$\alpha_s F_\varphi(\hat{\mathbf{x}}_\theta(\mathbf{z}_t, t), s) - \frac{\sqrt{\sigma_s^2 - \tilde{\sigma}_{s|t}^2}}{\sigma_t}\Big(\cancel{\mathbf{z}_t} - \alpha_t F_\varphi(\hat{\mathbf{x}}_\theta(\mathbf{z}_t, t), t)\Big)\Big\|_2^2 \tag{18}$$

$$= \frac{1}{2\tilde{\sigma}_{s|t}^2}\Big\|\alpha_s\Big(F_\varphi(\mathbf{x}, s) - F_\varphi(\hat{\mathbf{x}}_\theta(\mathbf{z}_t, t), s)\Big) +$$

$$\frac{\sqrt{\sigma_s^2 - \tilde{\sigma}_{s|t}^2}}{\sigma_t}\alpha_t\Big(F_\varphi(\hat{\mathbf{x}}_\theta(\mathbf{z}_t, t), t) - F_\varphi(\mathbf{x}, t)\Big)\Big\|_2^2. \tag{19}$$

With a learnable transformation $F_\varphi$, the term $\mathcal{L}_{\text{prior}}$ becomes dependent on the parameters $\varphi$, necessitating its optimization during training. We can compute the prior term as follows:

$$\text{D}_{\text{KL}}\Big(q_\varphi(\mathbf{z}_T|\mathbf{x})||p(\mathbf{z}_T)\Big) = \frac{1}{2}\left[\log\frac{|I|}{|\sigma_T^2 I|} - d + Tr\{I^{-1}\sigma_t^2 I\} + \left\|0 - \alpha_T F_\varphi(\mathbf{x}, T)\right\|_2^2\right] \quad (20)$$

$$= \frac{1}{2}\left[-d\log\sigma_T^2 - d + d\sigma_T^2 + \left\|\alpha_T F_\varphi(\mathbf{x}, T)\right\|_2^2\right] \quad (21)$$

$$= \frac{1}{2}\left[d\Big(\sigma_T^2 - \log\sigma_T^2 - 1\Big) + \alpha_T^2\left\|F_\varphi(\mathbf{x}, T)\right\|_2^2\right]. \quad (22)$$

Here, $d$ represents the dimensionality of the data space.

### A.3 REVERSE SDE AND ODE

As discussed in Section 3.2, when the number of steps, denoted as $T$, tends to infinity for NDM, we can switch to continuous time.

In the discrete time setting, we define the time step as $t \in [0, 1, \ldots, T]$. In the continuous time setting, we utilize the unit interval, denoting time as $t \in [0, 1]$. Nevertheless, for the sake of notational simplicity in this and subsequent sections, we will consider the discrete time to also lie within the unit interval, with $t \in [\frac{0}{T}, \frac{1}{T}, \ldots, \frac{T}{T}]$.

To derive the stochastic differential equation (SDE) for the reverse process $p_\theta(\mathbf{z}_s|\mathbf{z}_t)$ in NDM, we first obtain an SDE that depends on the data point $\mathbf{x}$ and whose solution corresponds to the posterior distribution $q_\varphi(\mathbf{z}_{t-\Delta t}|\mathbf{z}_t, \mathbf{x})$. By defining $p_\theta(\mathbf{z}_s|\mathbf{z}_t)$ through $q_\varphi(\mathbf{z}_s|\mathbf{z}_t, \mathbf{x})$ with the prediction $\hat{\mathbf{x}}_\theta(\mathbf{z}_t, t)$ instead of $\mathbf{x}$, we can subsequently replace the prediction $\hat{\mathbf{x}}_\theta(\mathbf{z}_t, t)$ and derive the SDE for the reverse process.

We constructively derive the SDE for the posteriors $q_\varphi(\mathbf{z}_s|\mathbf{z}_t, \mathbf{x})$. First, let us consider the following auxiliary SDE with backward time flow:

$$d\varepsilon_t = \frac{\dot{\nu}_t}{2}\varepsilon_t dt + \sqrt{\dot{\nu}_t}d\mathbf{w}. \quad (23)$$

It is straightforward to show that the solution to this SDE corresponds to the following distribution:

$$q(\varepsilon_s|\varepsilon_t) = \mathcal{N}(\varepsilon_s; \sqrt{1 - \bar{\sigma}_{s|t}^2}\varepsilon_t; \bar{\sigma}_{s|t}^2 I), \quad \text{where} \quad \bar{\sigma}_{s|t}^2 = 1 - e^{\nu_s - \nu_t}. \quad (24)$$

To derive the SDE for the posteriors $q_\varphi(\mathbf{z}_s|\mathbf{z}_t, \mathbf{x})$, we can apply the following function to both the SDE (23) and the distribution (24):

$$G(\mathbf{x}, \varepsilon_t, t) = \alpha_t F_\varphi(\mathbf{x}, t) + \sigma_t \varepsilon_t \quad (25)$$

Note that after applying the function $G$, the distribution (24) matches the posterior distribution $q_\varphi(\mathbf{z}_s|\mathbf{z}_t, \mathbf{x})$. Therefore, the desired SDE for $q_\varphi(\mathbf{z}_s|\mathbf{z}_t, \mathbf{x})$ is obtained by transforming the SDE (23) using Ito's formula (Øksendal & Øksendal, 2003):

$$d\mathbf{z}_t = \left[\frac{\partial G(\mathbf{x}, \varepsilon, t)}{\partial t}\bigg|_{\varepsilon=\varepsilon_t} + \frac{\dot{\nu}_t}{2}\frac{\partial G(\mathbf{x}, \varepsilon_t, t)}{\partial \varepsilon_t}\varepsilon_t - \frac{\dot{\nu}_t}{2}\frac{\partial^2 G(\mathbf{x}, \varepsilon_t, t)}{\partial \varepsilon_t^2}\right]dt + \sqrt{\dot{\nu}_t}\frac{\partial G(\mathbf{x}, \varepsilon_t, t)}{\partial \varepsilon_t}d\mathbf{w} \quad (26)$$

$$= \left[\dot{\alpha}_t F_\varphi(\mathbf{x}, t) + \alpha_t \dot{F}_\varphi(\mathbf{x}, t) + \dot{\sigma}_t\varepsilon_t + \frac{\dot{\nu}_t}{2}\sigma_t\varepsilon_t\right]dt + \sqrt{\dot{\nu}_t}\sigma_t d\mathbf{w} \quad (27)$$

$$= \left[\frac{\dot{\alpha}_t}{\alpha_t}(\mathbf{z}_t - \sigma_t\varepsilon_t) + \alpha_t\dot{F}_\varphi(\mathbf{x}, t) + \dot{\sigma}_t\varepsilon_t + \frac{\dot{\nu}_t}{2}\sigma_t\varepsilon_t\right]dt + \sqrt{\dot{\nu}_t}\sigma_t d\mathbf{w} \quad (28)$$

$$= \left[\alpha_t\dot{F}_\varphi(\mathbf{x}, t) + \frac{\partial\log\alpha_t}{\partial t}\mathbf{z}_t - \frac{1}{2}\left(\frac{\partial\sigma_t^2}{\partial t} - 2\frac{\partial\log\alpha_t}{\partial t}\sigma_t^2 + \dot{\nu}_t\sigma_t^2\right)\left(-\frac{\varepsilon_t}{\sigma_t}\right)\right]dt + \sqrt{\dot{\nu}_t}\sigma_t d\mathbf{w} \quad (29)$$

$$= \left[\alpha_t\dot{F}_\varphi(\mathbf{x}, t) + r(t)\mathbf{z}_t - \frac{1}{2}\left(\frac{\partial\sigma_t^2}{\partial t} - 2r(t)\sigma_t^2 + \dot{\nu}_t\sigma_t^2\right)s(\mathbf{x}, \mathbf{z}_t, t)\right]dt + \sqrt{\dot{\nu}_t}\sigma_t d\mathbf{w}, \quad (30)$$

$$\text{where} \quad r(t) = \frac{\partial\log\alpha_t}{\partial t} \quad \text{and} \quad s(\mathbf{x}, \mathbf{z}_t, t) = \frac{\alpha_t F_\varphi(\mathbf{x}, t) - \mathbf{z}_t}{\sigma_t^2} \quad (31)$$

To obtain the SDE for the reverse process, we can substitute the prediction $\hat{\mathbf{x}}_\theta(\mathbf{z}_t, t)$ instead of $\mathbf{x}$. This substitution yields the SDE (13):

$$d\mathbf{z}_t = \left[\alpha_t \dot{F}_\varphi(\hat{\mathbf{x}}_\theta(\mathbf{z}_t, t), t) + r(t)\mathbf{z}_t - \frac{1}{2}\left(\frac{\partial \sigma_t^2}{\partial t} - 2r(t)\sigma_t^2 + \dot{\nu}_t \sigma_t^2\right) s_\theta(\mathbf{z}_t, t)\right] dt + \sqrt{\dot{\nu}_t}\sigma_t d\mathbf{w},$$

(32)

$$\text{where} \quad s_\theta(\mathbf{z}_t, t) = \frac{\alpha_t F_\varphi(\hat{\mathbf{x}}_\theta(\mathbf{z}_t, t), t) - \mathbf{z}_t}{\sigma_t^2}$$

(33)

As discussed earlier, we can leverage the Jacobian-Vector Product trick (Hirsch et al., 2012) to calculate $\dot{F}_\varphi$.

In the case where $\nu_t$ is a constant, the dynamics become deterministic and can be described by ordinary differential equations (ODEs). In our experiments, we utilize these ODEs to model the generative process as a continuous normalizing flow (Chen et al., 2018; Grathwohl et al., 2018) and estimate densities.

## A.4 CONTINUOUS TIME OBJECTIVE

When we switch to continuous time, the discrete objective (10) transforms from finite sum of KL divergances into integral, which we can easily derive as soon as we have access to both stochastic differential equations associated with the forward process (30) and with the rewerse process (32). In continuous time the diffusion term $\mathcal{L}_{\text{diff}}$ (11) is equal to:

$$\mathcal{L}_{\text{diff}} = \int_0^1 \frac{1}{g^2(t)}\left\|\alpha_t\left(\dot{F}_\varphi(\mathbf{x}, t) - \dot{F}_\varphi(\hat{\mathbf{x}}_\theta(\mathbf{z}_t, t), t)\right) + \right.$$

$$\left.\frac{1}{2}\left(\frac{\partial \sigma_t^2}{\partial t} - 2r(t)\sigma_t^2 + g^2(t)\right)\left(s(\mathbf{x}, \mathbf{z}_t, t) - s(\hat{\mathbf{x}}_\theta(\mathbf{z}_t, t), \mathbf{z}_t, t)\right)\right\|_2^2 dt,$$

$$\text{where} \quad r(t) = \frac{\partial \log \alpha_t}{\partial t}, \quad g^2(t) = \dot{\nu}_t \sigma_t^2 \quad \text{and} \quad s(\mathbf{x}, \mathbf{z}_t, t) = \frac{\alpha_t F_\varphi(\mathbf{x}, t) - \mathbf{z}_t}{\sigma_t^2}.$$

(34)

As we can see, these equation contains $\dot{F}_\varphi$ as a component. In general, we do not have explicit access to the time derivative of the forward transformation $F_\varphi$. However, we will focus on cases where the forward transformation is differentiable. By utilizing automatic differentiation tools, we can calculate the time derivatives of $F_\varphi$. Nevertheless, when $\mathbf{x}$ is fixed, the function $F_\varphi(\mathbf{x}, \cdot)$ becomes a scalar-to-vector function. To compute its time derivative using simple backpropagation, we would need to execute it for all outputs of $F_\varphi$, resulting in quadratic computational complexity. Fortunately, there exists a more efficient method to obtain the time derivative, the Jacobian-Vector Product trick (Hirsch et al., 2012). The Jacobian of the transformation function with $\mathbf{x}$ fixed is represented as a column matrix. Therefore, by computing the product of the Jacobian with a one-dimensional vector, we can obtain a vector of time derivatives.

## B CONNECTIONS WITH OTHER WORKS

We introduce NDMs as a comprehensive framework that generalises various existing approaches. Here we provide Table 5 which is an extended version of Table 1, that demonstrates how existing approaches appear as a spatial cases of NDMs.

Here we provide an extended discussion on the connection between NDM and other related works.

### B.1 DIFFUSION IN LATENT SPACE

The concept of a learnable forward process is not entirely new. In some sense models that run a diffusion process in the latent space of a VAE (Vahdat et al., 2021; Rombach et al., 2022), a hierarchical VAE (Gu et al., 2022), or a Flow model (Kim et al., 2022) can be viewed as diffusion

Table 5: Summary of existing diffusion models as instances of Neural Diffusion Models (NDM).

| Model | Distribution $q(\mathbf{z}_t\|x)$ | NDM's $F(\mathbf{x}, t)$ | Comment |
|---|---|---|---|
| DDPM (Ho et al., 2020) / DDIM (Song et al., 2020a) | $\mathcal{N}\big(\mathbf{z}_t; \alpha_t\mathbf{x}, \sigma_t^2 I\big)$ | $\mathbf{x}$ | |
| Flow Matching OT (Lipman et al., 2022) | $\mathcal{N}\big(\mathbf{z}_t; \alpha_t\mathbf{x}, \sigma_t^2 I\big)$ | $\mathbf{x}$ | $\alpha_t = t,$ $\sigma_t = 1 - (1 - \sigma_{\min})t$ |
| VDM (Kingma et al., 2021) | $\mathcal{N}\big(\mathbf{z}_t; \alpha_t\mathbf{x}, \sigma_t^2 I\big)$ | $\mathbf{x}$ | $\alpha_t^2 = \mathrm{sigmoid}(-\gamma_\eta(t)),$ $\sigma_t^2 = \mathrm{sigmoid}(\gamma_\eta(t))$ |
| IHDM (Rissanen et al., 2022) | $\mathcal{N}\big(\mathbf{z}_t; Ve^{-\Lambda t}V^T\mathbf{x}, \sigma^2 I\big)$ | $Ve^{-\Lambda t}V^T\mathbf{x}$ | $\alpha_t = 1, \sigma_t = \sigma,$ $\sigma$ is fixed |
| Blurring Diffusion (Hoogeboom & Salimans, 2022) | $\mathcal{N}\big(\mathbf{z}_t; \alpha_t e^{-\Lambda t}V^T\mathbf{x}, \sigma_t^2 I\big)$ | $e^{-\Lambda t}V^T\mathbf{x}$ | $p(x\|z_0) = \mathcal{N}\big(x; aVz_0, \sigma\big)$ |
| Soft Diffusion (Daras et al., 2022) | $\mathcal{N}\big(\mathbf{z}_t; C_t\mathbf{x}, s_t^2 I\big)$ | $C_t\mathbf{x}$ | $\alpha_t = 1, \sigma_t^2 = s_t^2$ |
| LSGM (Vahdat et al., 2021) | $\mathcal{N}\big(\mathbf{z}_t; \alpha_t E(\mathbf{x}), \sigma_t^2 I\big)$ | $E(\mathbf{x})$ | $p(x\|z_0) = \mathcal{N}\big(x; aD(z_0), \sigma^2\big)$ |
| f-DM (Gu et al., 2022) | $\mathcal{N}\big(\mathbf{z}_t; \alpha_t\mathbf{x}_t, \sigma_t^2 I\big)$ | $\mathbf{x}_t = \frac{(t-\tau_k)\hat{x}^k + (\tau_{k+1}-t)x^k}{\tau_{k+1}-\tau_k},$ where $\tau_k \leq t < \tau_{k+1}$ | $x^k = f_{0:k}(x)$ $\hat{x}^k = \begin{cases} g_k(f_{k+1}(x^k)), & \text{if } k < K, \\ x^k, & \text{if } k = K. \end{cases}$ |

models with a learnable forward process. These models optimize the mapping to the latent space. Consequently, projecting the diffusion generative dynamic from the latent to the data space introduces a novel, nonlinear, and learnable generative dynamic. However, these models still rely on conventional diffusion in the latent space.

Additionally, these models can be viewed as spatial cases of NDM with a specific choice of the transformation $F_\varphi(\mathbf{x}, t)$. For example, $F_\varphi$ might be selected as the VAE's time independent encoder in the case of (Vahdat et al., 2021) or the time independent Flow model in the case of (Kim et al., 2022).

## B.2 STOCHASTIC INTERPOLANTS

Albergo & Vanden-Eijnden (2022) proposed a Stochastic Interpolant approach, which provide more flexibility then conventional diffusion models in defining and even learning of the forward process. While we find stochastic interpolants intriguing and promising, as well as related to our work, these methods differ significantly.

Firstly, stochastic interpolants represent an approach to learning continuous-time deterministic generative dynamics, whereas NDM learns stochastic dynamics in either discrete or continuous time, which can subsequently may be transformed into a deterministic process.

Secondly, in NDM, the model is trained by optimizing the variational bound on the likelihood, while Stochastic Interpolants are trained by optimizing the generalization of the Flow Matching objective (Lipman et al., 2022).

Lastly, NDM joint learns both the forward and reverse processes by optimizing the likelihood, whereas stochastic interpolants learn the generative process with a fixed forward process. Albergo & Vanden-Eijnden (2022) demonstrate the possibility of constructing an optimization procedure for the forward process through a max-min game to solve a dynamic optimal transport problem. However, the purpose of this optimization differs from that of NDM.

Moreover, max-min optimization, as employed in Stochastic Interpolants, is notably less stable compared to min-min optimization in NDM. Additionally, Stochastic Interpolants do not present experimental results for the optimization of the forward process.

## B.3 SCHRÖDINGER BRIDGES

Another line of works (De Bortoli et al., 2021; Wang et al., 2021; Peluchetti; Chen et al., 2021) are approaches based on Schrödinger Bridge theory. While such approaches allow learning forward transformations, in contrast to NDM, these approaches are not simulation-free. In Schrödinger Bridge

models, we typically lack direct access to the distribution $q(\mathbf{z}_t|\mathbf{x})$. Consequently, to sample the latent variable $\mathbf{z}_t$ in training time, we must simulate the full stochastic process, such as the stochastic differential equations. This characteristic makes Schrödinger Bridge models expensive in training and not simulation-free.

In contrast, NDM framework, by design, has access to $q(\mathbf{z}_t|\mathbf{x})$. Thus, with NDM, when training a model with $T$ time steps, there is no need to propagate $F_\varphi$ for $T$ times at each step of the training procedure. Instead, the NDM framework enables sampling of the intermediate latent variables $\mathbf{z}_t$ directly from the distribution $q(\mathbf{z}_t|\mathbf{x})$. Therefore, we can maintain the training paradigm outlined in Section 2. Instead of computing all $T$ KL divergences for each time step, we can approximate the objective using the Monte Carlo method by calculating just one KL divergence for a uniformly sampled time step $t \in [1; T]$, as described in Algorithm 1.

This approach allows us to train the model with batches of shape $[batch\_size, d]$ rather than $[batch\_size, T, d]$. Consequently, NDM can leverage larger batch sizes and use just one call of $F_\varphi$ for inferring latent variables $\mathbf{z}_t$.

### B.4 DIFFENC

In concurrent work, published to arXiv after the submission deadline, Nielsen et al. (2023) introduced DiffEnc. DiffEnc also proposes to add a time-dependent transformation to the data in the diffusion model. However, there are some distinctions between these two methods. Firstly, in NDM, we parameterize the reverse process by predicting the data point $\mathbf{x}$, while in DiffEnc, they predict the transformed data point $F_\varphi(\mathbf{x}, t)$.

Secondly, in NDM, we employ a Signal-to-Noise Ratio (SNR) schedule for noise injection from DDPM (Ho et al., 2020) and a straightforward parameterization of the model $\hat{\mathbf{x}}_\theta(\mathbf{z}_t, t)$ through predicting the injected epsilon, as detailed in Appendix C.2. Simultaneously, in DiffEnc, the authors use a learnable SNR schedule (Kingma et al., 2021) and a v-parameterization (Salimans & Ho, 2022) of $\hat{\mathbf{x}}_\theta(\mathbf{z}_t, t)$.

Finally, DiffEnc utilizes approximations of the time derivatives of data transformations $F_\varphi$, while in the NDM framework, we propose calculating exact time derivatives using Jacobian-Vector Products.

### B.5 DISCUSSION

In theory, we can view the diffusion model as a hierarchical VAE. From this perspective, the conventional diffusion model can be seen as a VAE with a fixed variational distribution. In contrast NDMs with learnable transformations have the capability to make this distribution learnable. Consequently, it effectively reduces the gap between the log likelihood and the variational bound.

In practical terms, when dealing with reverse processes parameterised by neural networks, which inherently possess their own biases, the learnable $F_\varphi$ function tends to acquire a transformation that facilitates the fitting of the reverse process to the forward process. Therefore, as soon as there are infinitely many pairs of forward and reverse processes with same joint distributions of latent variables, it's hard to say what $F_\varphi$ should be in general case. It may learn anything that fits to the reverse process. However, besides better likelihood estimation, NDMs with learnable transformations gives us additional flexibility in how we can define the reverse process.

In Appendix E we provide a proof of concept experiment, which demonstrates that we can learn simpler generative dynamics compared to conventional diffusion models. In this experiment we restrict the reverse process to learn dynamic optimal transport trajectories only, and learn forward and reverse processes end-to-end. It is not possible to match such a reverse process with a predefined forward process, but NDMs allows to capture the data distribution with the simpler generative dynamics.

## C IMPLEMENTATION DETAILS

All our experiments were conducted using synthetic 2D datasets and image datasets: MNIST (Deng, 2012), CIFAR-10 (Krizhevsky et al., 2009), downsampled ImageNet (Deng et al., 2009; Chrabaszcz et al., 2017) and CelebA-HQ-256 (Karras et al., 2017). For CIFAR-10 and ImageNet datasets we

Table 6: Training hyper-parameters.

|  | CIFAR-10 | ImageNet 32 | ImageNet 64 |
|---|---|---|---|
| Channels | 256 | 256 | 192 |
| Depth | 2 | 3 | 3 |
| Channels multipliers | 1,2,2,2 | 1,2,2,2 | 1,2,3,4 |
| Heads | 4 | 4 | 4 |
| Heads Channels | 64 | 64 | 64 |
| Attention resolution | 16 | 16,8 | 32,16,8 |
| Dropout | 0.0 | 0.0 | 0.0 |
| Effective Batch size | 256 | 1024 | 2048 |
| GPUs | 2 | 4 | 16 |
| Epochs | 1000 | 200 | 250 |
| Iterations | 391k | 250k | 157k |
| Learning Rate | 4e-4 | 1e-4 | 1e-4 |
| Learning Rate Scheduler | Polynomial | Polynomial | Constant |
| Warmup Steps | 45k | 20k | - |

applied center cropping and resizing, following the same pre-processing steps as Chrabaszcz et al. (2017). For synthetic data, we employed a 5-layer MLP with 512 neurons in each layer, while for the images, we utilized the U-Net architecture from Dhariwal & Nichol (2021). In our experiments both the DDPM and NDM approaches were trained on identical architectures, with the same hyper-parameters and for the same number of epochs. The hyper-parameters are presented in Table 6. In experiment where we report results for the continuous time models we use importance sampling of time (Song et al., 2021) instead of uniform sampling.

We trained models using the Adam optimizer, setting the following parameters: $\beta_1 = 0.9$, $\beta_2 = 0.999$, weight decay of 0.0, and $\varepsilon = 10^{-8}$. To facilitate the training process, we employed a polynomial decay learning rate schedule, which includes a warm-up phase for a specified number of training steps. During the warm-up phase, the learning rate is linearly increased from $10^{-8}$ to the peak learning rate. Once the peak learning rate is reached, the learning rate is linearly decayed to $10^{-8}$ until the final training step. The training was performed using Tesla V100 GPUs.

## C.1 DEQUANTIZATION

When reporting negative log-likelihood, we dequantize using the standard uniform dequantization. We report an importance-weighted estimate using

$$\log \frac{1}{K} \sum_{k=1}^{K} p_\theta(\mathbf{x} + u_k), \quad \text{where} \quad u_k \sim \mathcal{U}(0,1), \tag{35}$$

with $\mathbf{x} \in [0, \ldots, 255]$.

## C.2 PARAMETERIZATION

In order to simplify the derivations above, we have utilized the notation $\hat{\mathbf{x}}_\theta(\mathbf{z}_t, t)$ to represent the prediction of the reverse process. However, prior research has shown that predicting the injected noise $\varepsilon_t$ can lead to improved results (Ho et al., 2020; Nichol & Dhariwal, 2021; Dhariwal & Nichol, 2021). Therefore, in all the experiments, we opt for the following parameterization:

$$\hat{\mathbf{x}}_\theta(\mathbf{z}_t, t) = \frac{\mathbf{z}_t - \sigma_t \hat{\varepsilon}_\theta(\mathbf{z}_t, t)}{\alpha_t}. \tag{36}$$

It is worth noting that with this parameterization, $\hat{\varepsilon}_\theta(\mathbf{z}_t, t)$ does not necessarily approximate the true injected noise $\varepsilon_t$, since this reparameterization does not account for the transformation $F_\varphi$. We believe that better parameterizations may exist for NDM, but we leave this for future research.

Furthermore, we restrict the transformation $F_\varphi$ to an identity transformation for $t = 0$ through the following construction:

$$F_\varphi(\mathbf{x}, t) = (1 - t)\mathbf{x} + t\bar{F}_\varphi(\mathbf{x}, t). \tag{37}$$

Table 7: Performance comparison the DDPM and NDM on CIFAR-10 and ImageNet 32 datasets with different numbers of steps. We report the performance with same hyperparameters and neural networks on both models to quantify the effect of learnable transformation in fair setting. We provide likelihood (bits/dim) and negative ELBO. Additionally for CIFAR-10 and ImageNet 32 we provide FID score. Boldface numbers represent the best performance. NDM consistently outperforms in terms of NLL and NELBO with comparable sample quality to DDPM on all datasets.

| Steps | Model | CIFAR-10 | | | ImageNet 32 | | |
|---|---|---|---|---|---|---|---|
| | | NLL ↓ | NELBO ↓ | FID ↓ | NLL ↓ | NELBO ↓ | FID ↓ |
| 1000 | DDPM | 3.11 | 3.18 | **11.44** | 3.89 | 3.95 | **16.18** |
| | NDM | **3.02** | **3.03** | 11.82 | **3.79** | **3.82** | 17.02 |
| 100 | DDPM | 3.31 | 3.38 | **11.78** | 4.14 | 4.23 | **16.66** |
| | NDM | **3.05** | **3.12** | 11.98 | **3.83** | **3.92** | 17.74 |
| 50 | DDPM | 3.49 | 3.57 | 13.22 | 4.37 | 4.47 | **18.70** |
| | NDM | **3.22** | **3.30** | **13.15** | **4.05** | **4.14** | 18.93 |
| 10 | DDPM | 5.02 | 5.13 | 37.83 | 6.28 | 6.42 | 53.51 |
| | NDM | **4.63** | **4.74** | **31.56** | **5.81** | **5.94** | **45.38** |
| 1000 → 100 | DDPM | 3.38 | 3.45 | **12.29** | 4.23 | 4.32 | **17.49** |
| | NDM | **3.30** | **3.37** | 12.70 | **4.15** | **4.23** | 18.48 |
| 1000 → 50 | DDPM | 4.08 | 4.17 | **15.24** | 5.10 | 5.21 | **20.09** |
| | NDM | **3.98** | **4.07** | 16.83 | **5.00** | **5.10** | 21.11 |
| 1000 → 10 | DDPM | 8.78 | 8.98 | **43.85** | 10.99 | 11.23 | **58.35** |
| | NDM | **8.58** | **8.81** | 48.41 | **10.78** | **11.06** | 62.12 |

This ensures that $q(\mathbf{z}_0|\mathbf{x}) \approx \delta(\mathbf{z}_0 - \mathbf{x})$, and thus also removes the need to optimize the reconstruction term $\mathcal{L}_{\text{rec}}$.

Finally, to ensure consistency with Ho et al. (2020) we use $\tilde{\sigma}^2_{s|t} = \left( \sigma^2_t - \frac{\alpha^2_t}{\alpha^2_s} \sigma^2_s \right) \frac{\sigma^2_s}{\sigma^2_t}$ for the forward process (8). This choice of $\tilde{\sigma}^2_{s|t}$ guaranties consistency between the NDM and DDPM forward processes. For $\alpha_t$ and $\sigma^2_t$ we use the DDPM schedule of noise injection.

### C.3 DIFFUSION IN LATENT SPACE

For experiment with diffusion in the latent space of VAE on CelebA-HQ-256, we followed LSGM (Vahdat et al., 2021) experiment setup. The only difference between LSGM baseline and our model is that we utilize learnable transformations $F_\varphi$ according to NDMs framework. We apply the same hyperparameters, as LSGM.

## D  ADDITIONAL RESULTS

### D.1  ADDITIONAL EVALUATION

Here we provide Table 7 which contains additional resalts to Table 4. This table compare DDPM and NDMs with learnable transformations on CIFAR-10 and ImageNet $32 \times 32$ datasets with different numbers of steps.

### D.2  ADDITIONAL SAMPLES

In this section, we present additional illustrations showcasing the properties of NDMs.

Figure 4 provides a comparison between DDPM and NDM on a synthetic 2D data distribution. For this experiment, both models utilize $T = 10$ discrete time steps. From Figure 4c, it is evident that NDM learns to transform the data distribution. Additionally, after injecting noise (Figures 4a and 4d),

Table 8: Comparison of NDM and DDPM with doubled number of parameters on CIFAR-10 for 10 and 1000 steps. The performance of DDPM stays the same while doubling the number of parameters, and NDM still achieves the best NLL and NELBO despite comparable number of parameters.

| | 10 steps | | | 1000 steps | | |
|---|---|---|---|---|---|---|
| Model | NLL ↓ | NELBO ↓ | FID ↓ | NLL ↓ | NELBO ↓ | FID ↓ |
| DDPM | 5.02 | 5.13 | 37.83 | 3.11 | 3.18 | 11.44 |
| DDPM (stack) | 5.02 | 5.13 | 38.05 | 3.10 | 3.18 | 11.42 |
| DDPM (wide) | 5.01 | 5.11 | 37.88 | 3.11 | 3.17 | **11.39** |
| NDM | **4.63** | **4.74** | **31.56** | **3.02** | **3.03** | 11.82 |

the distributions of samples $\mathbf{z}_t$ show minimal differences between DDPM and NDM. However, when examining the predictions of data points $\hat{\mathbf{x}}_\theta(\mathbf{z}_t, t)$ (Figures 4b and 4e), NDM produces predictions that more closely resemble the true data distribution compared to DDPM.

A similar pattern emerges when applying these models to the MNIST dataset, as depicted in Figure 5. For this experiment we also use $T = 10$ discrete time steps. DDPM generates blurry predictions $\hat{\mathbf{x}}_\theta(\mathbf{z}_t, t)$ for $t$ close to $T$, which bear little resemblance to real MNIST samples. Conversely, NDM produces predictions that are more similar to the true MNIST distribution, despite both models generating similar-looking noisy samples.

Finally, we include samples from both DDPM and NDM models with $T = 1000$ steps on the CIFAR-10 dataset in Figure 6. As outlined in Table 1, NDM exhibits lower sample quality based on FID measurements; however, visually there is no drop in quality.

### D.3 ABLATION STUDIES

Finally, we address the question of whether the improved performance of NDM is due to the proposed method or merely the result of increasing the number of model parameters. To investigate this issue, we provide additional experiments where we double the number of DDPM parameters in two ways. The first way is to simply stack two U-Net architectures, which is the closest form to NDM. The second way is to increase the width of the U-Net architecture. Specifically, for the second way we use 384 channels instead of 256. Importantly, we left all other hyper-parameters (see Table 6), such as the learning rate and number of iterations, unchanged. As shown in Table 8, neither of these approaches yields the same results as NDM with learnable transformations. This means that the improved performance is not simply a result of the increased number of parameters.

## E DYNAMIC OPTIMAL TRANSPORT

In this section, we present a proof-of-concept experiment demonstrating that the NDMs framework enables the learning of simpler generative trajectories. Specifically, we conduct experiments involving a 1D mixture of Gaussian distribution and dynamic optimal transport (OT).

While NDMs don't inherently have a direct connection with OT, we can establish a connection given the presence of infinitely many pairs of matched forward and reverse processes. This connection is facilitated by the NDMs' ability to learn the forward process. Therefore, we can consider the following setup.

We consider NDMs with a learnable function $F_\varphi$. Then, we constrain the reverse process to exclusively learn dynamic OT mappings. Finally, we train both the forward and reverse processes jointly, following the NDMs framework. In such a setup we can expect the forward process to learn such a transition from data distribution to Gaussian distribution, that aligns with the limitations imposed on the reverse process.

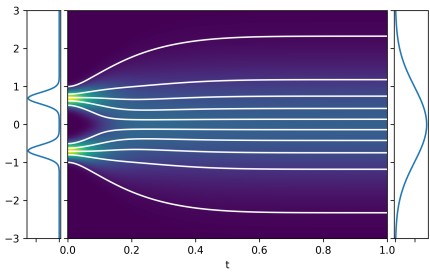

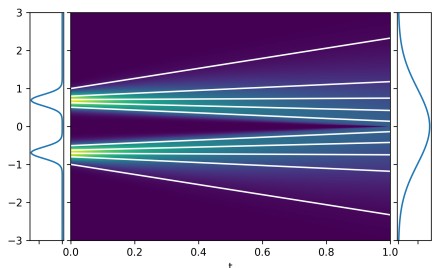

(a) DDPM with regular reverse process.

(b) NDM with restricted (OT) reverse process.

Figure 3: Comparison of DDPM and NDM with restricted reverse process to be optimal transport, 1D distribution.

### E.1 RESTRICTED REVERSE PROCESS

To restrict the reverse process we parameterise the reverse deterministic process to have linear trajectories:

$$\mathbf{z}_t = h_\theta(t, \varepsilon) = (1 - t)\hat{\mathbf{x}}_\theta(\varepsilon) + t\varepsilon, \tag{38}$$

where $\varepsilon$ is a sample drawn from a unit Gaussian distribution. Since we are working with smooth 1D distributions, it is enough for $\hat{\mathbf{x}}_\theta$ to be monotonically increasing, so the trajectories $\mathbf{z}_t$ correspond to dynamic OT. Which means that for any parameters $\theta$ the reverse process describes dynamic OT between the standard Gaussian distribution and another distribution (not necessarily exactly the target data distribution). In practice, we parameterize $\hat{\mathbf{x}}_\theta$ using the neural network proposed by Kingma et al. (2021) for the parameterization of the Signal-to-Noise Ratio (SNR) function.

Then, we can derive an ordinary differential equation (ODE) for the reverse process:

$$d\mathbf{z}_t = \underbrace{\varepsilon - \hat{\mathbf{x}}_\theta(\varepsilon)\Big|_{\varepsilon=h_\theta^{-1}(t,\mathbf{z}_t)}}_{f_\theta(t,\mathbf{z}_t)} dt. \tag{39}$$

Next, we may switch to a stochastic differential equation (SDE) according to Song et al. (2020b):

$$d\mathbf{z}_t = \underbrace{\left[ f_\theta(t, \mathbf{z}_t) - \frac{g^2(t)}{2} \nabla_{\mathbf{z}_t} \log p_\theta(\mathbf{z}_t) \right]}_{f_\theta^r(t,\mathbf{z}_t)} dt + g(t)d\bar{\mathbf{w}}. \tag{40}$$

As soon as we have access to $h_\theta^{-1}$, we may find:

$$\nabla_{\mathbf{z}_t} \log p_\theta(\mathbf{z}_t) = \nabla_{\mathbf{z}_t} \left[ \log p(\varepsilon) - \log \left| \frac{\partial \mathbf{z}_t}{\partial \varepsilon} \right| \right] \Big|_{\varepsilon=h_\theta^{-1}(t,\mathbf{z}_t)} \tag{41}$$

$$= \nabla_{\mathbf{z}_t} \left[ \log p(\varepsilon) - \log \left| (1-t)\frac{\partial \mathbf{x}_t}{\partial \varepsilon} + t \right| \right] \Big|_{\varepsilon=h_\theta^{-1}(t,\mathbf{z}_t)}. \tag{42}$$

### E.2 OBJECTIVE FUNCTION

To train a model with such a specific reverse process, we can utilize a slightly modified NDMs framework. The only component of the NDMs' objective that is unclear is the diffusion term $\mathcal{L}_{\text{diff}}$. NDMs provide a conditional reverse SDE associated with the forward process (30) in the following form:

$$d\mathbf{z}_t = f_\varphi^f(x, t, \mathbf{z}_t)dt + g(t)d\bar{\mathbf{w}}. \tag{43}$$

Also, here we have the reverse SDE (40). Therefore, we may find diffusion term $\mathcal{L}_{\text{diff}}$ of objective as follows:

$$\mathcal{L}_{\text{diff}} = \mathbb{E}_{q(\mathbf{x})}\mathbb{E}_{u(t)}\mathbb{E}_{q(\mathbf{z}_t|\mathbf{x})}\frac{1}{g^2(t)}\left\|f_\varphi^f(x,t,\mathbf{z}_t) - f_\theta^r(t,\mathbf{z}_t)\right\|_2^2. \qquad (44)$$

### E.3  RESULTS AND DISCUSSION

Figures 3a and 3b illustrate trajectories learned by DDPM and NDM with learnable $F_\varphi$ and restricted reverse process. As expected, DDPM learns curved trajectories predetermined by fixed forward process. At the same time NDM effectively learns dynamic OT. It worths noting that DDPM with the restricted reverse process is by design not able to learn the data distribution, since it's impossible to match the fixed forward process (with curved trajectories) with the reverse process (with straight trajectories).

The proposed approach is limited to 1D data, monotonically increasing $\hat{\mathbf{x}}_\theta$, and a nontrivial $h_\theta^{-1}$ function, which we resolve using 5 iterations of Newton's method. Nevertheless, this experiment clearly demonstrates that NDMs may be utilised for learning OT as well as other (e.g. computationally efficient ones) dynamics by restricting the reverse process. Establishing rigorous theoretical connections with OT, developing specific techniques for efficient parameterisation of the reverse process and generalising to higher dimensions are interesting avenue for future work.

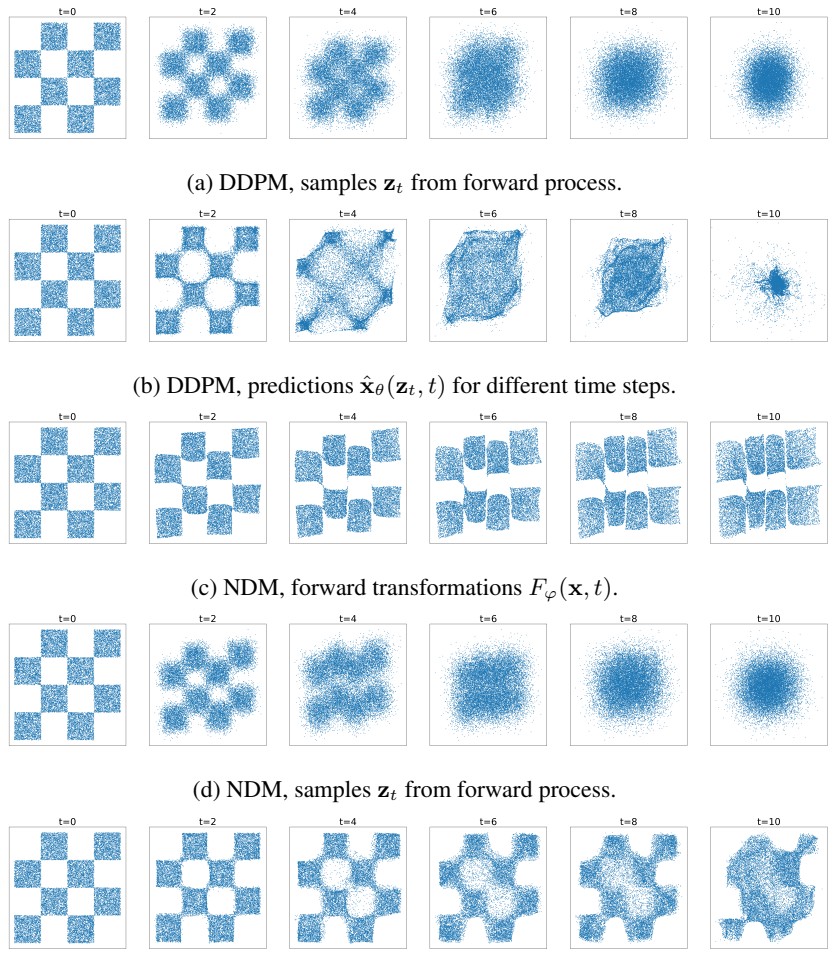

(a) DDPM, samples $\mathbf{z}_t$ from forward process.

(b) DDPM, predictions $\hat{\mathbf{x}}_\theta(\mathbf{z}_t, t)$ for different time steps.

(c) NDM, forward transformations $F_\varphi(\mathbf{x}, t)$.

(d) NDM, samples $\mathbf{z}_t$ from forward process.

(e) NDM, predictions $\hat{\mathbf{x}}_\theta(\mathbf{z}_t, t)$ for different time steps.

Figure 4: Comparison of DDPM and NDM on 2D distribution.

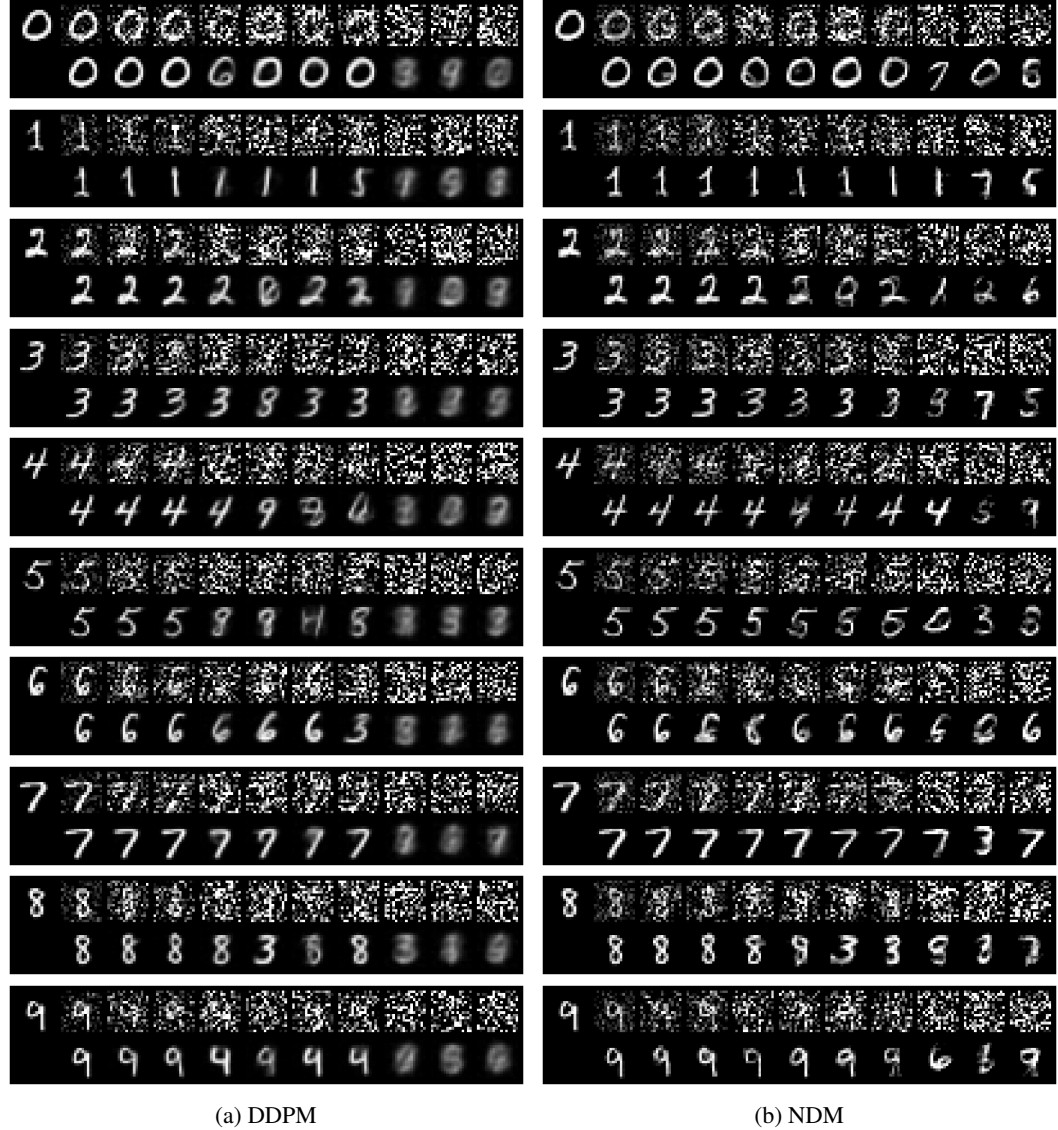

(a) DDPM  (b) NDM

Figure 5: Samples $\mathbf{z}_t$ from forward process and predicted data points $\hat{\mathbf{x}}_\theta(\mathbf{z}_t, t)$ on MNIST. (a) Samples from DDPM. (b) Samples from NDM. In each group, *Left:* data sample, *Top:* noised samples $\mathbf{z}_t$, *Bottom:* predicted data points $\hat{\mathbf{x}}_\theta(\mathbf{z}_t, t)$.

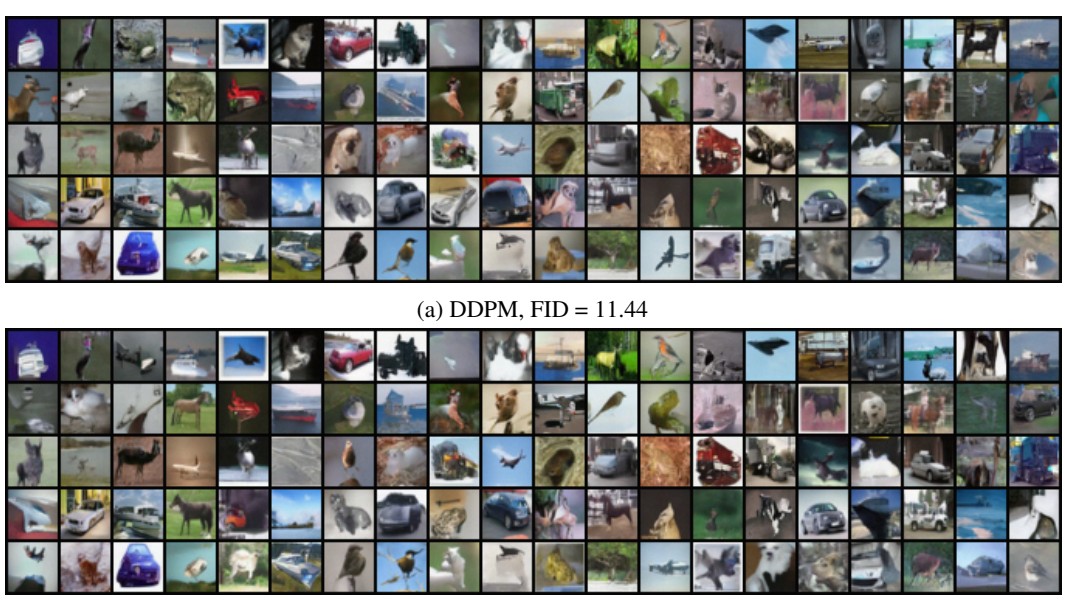

(a) DDPM, FID = 11.44

(b) NDM, FID = 11.82

Figure 6: Samples on CIFAR-10. (a) Samples from DDPM. (b) Samples from NDM. Samples of both models are generated with the same random seed.