# OpenReview forum: "Neural Diffusion Models"
_ICLR.cc/2024/Conference — Submitted to ICLR 2024_

### Official Review · Reviewer_vgew · 2023-10-20

**Soundness:** 2 fair
**Presentation:** 3 good
**Contribution:** 2 fair
**Rating:** 5
**Confidence:** 3

**Summary:**

This paper introduces Neural Diffusion Models (NDMs) as an extension of traditional diffusion models. While conventional diffusion models are limited to linear data transformations, NDMs allow for time-dependent non-linear transformations, potentially improving generative distribution training efficiency. The authors propose a variational bound for optimizing NDMs in a simulation-free setting and develop a time-continuous formulation for efficient inference. Experimental results on image generation benchmarks, such as CIFAR-10, downsized ImageNet, and CelebA-HQ, demonstrate that NDM is able to do generative tasks.

**Strengths:**

The paper introduces NDM as a framework that extends conventional diffusion models to both discrete and continuous time settings. The upper bound of the negative log-likelihood objective is provided. The authors claim that the generation quality is improved with small to medium steps in terms of log-likelihood. The approach is interesting, even though the idea of using a nonlinear forward process is not novel.

**Weaknesses:**

1. There are a few other learnable forward process which generalizes the diffusion model [1,2]. The authors should consider citing or comparing with them.

2. Actually I am not quite convincing by the effectiveness of proposed method. The numerical value is compared with the DDPM. The authors clearly present the differences with DDIM in Figure 1, but the experiment section is just focus on DDPM for fast sampling. The numerical value provided by authors is not impressive or competitive compared with the other fast sampling techniques on the market which is built on DDPM/DDIM [3,4].

[1]:Dongjun Kim et al. 'Maximum Likelihood Training of Implicit Nonlinear Diffusion Models.'

[2]: Tianrong Chan et al. 'Likelihood Training of Schrödinger Bridge using Forward-Backward SDEs Theory.'

[3]: Qinsheng Zhang et al. 'Fast Sampling of Diffusion Models with Exponential Integrator'

[4]: Fan Bao et al. 'Analytic-DPM'

**Questions:**

1. What is the difference between the NDM compared with [1,2]? What is the benefits over them?
2. In the experiment section, 'DDPM' represents for the SDE/stochastic model? If it is, it would be great to have the comparison for ODE model which is more favorable for fast sampling for both NDM and DDIM.
3. How do the authors obtain the $\hat{x}_{\theta}$ in eq.10? Does it include the inference of the network? If yes, then why the algorithm is simulation-free? From my understanding, it is implicitly simulating the dynamics which is also the part of reason for heavy training complexity as stated by the authors.

[1]:Dongjun Kim et al. 'Maximum Likelihood Training of Implicit Nonlinear Diffusion Models.'

[2]: Tianrong Chan et al. 'Likelihood Training of Schrödinger Bridge using Forward-Backward SDEs Theory.'

---

> ### Author Response · Authors · 2023-11-20
>
> We thank the reviewer for their review and suggestions which will help improve the paper. We will include the suggested reference in the revision. Below we address the questions and comments raised in the review.
>
> **Weaknesses and Questions:**
>
> 1. We present NDM not merely as a model learning the forward process but as a comprehensive framework that enables the predefined specification of forward dynamics capable of information destruction. Additionally, it facilitates the learning of these dynamics. The concept of a learnable forward process is not novel. However, unlike others, NDM allows time-dependent, non-linear, and learnable data transformations, all while remaining a simulation-free framework, which is an essential factor for the efficiency of the training procedure.
>
>     For instance, in [1], the authors suggest training a conventional diffusion model in the latent space of a flow model. This model can be viewed as a special case of NDM, where the transformation $F^\varphi$ corresponds to a time-independent flow.
>
>     The work [2], like many other Schrödinger Bridge models, does not fall into the special case of NDM. However, in contrast to NDM, the approach [2] is not simulation-free. In Schrödinger Bridge models, we typically lack access to the distribution $q(\mathbf{z}^t|\mathbf{x})$. Consequently, to sample the latent variable $\mathbf{z}^t$ in training time, we must simulate the full stochastic process, such as the SDE in [2]. This characteristic makes Schrödinger Bridge models expensive in training and not simulation-free. In contrast, NDM, by design, has access to $q(\mathbf{z}^t|\mathbf{x})$. Thus, with NDM, we can directly sample $\mathbf{z}^t$ without expending computational resources on simulation, enhancing the efficiency of the training procedure.
>
> 2. While NDM can be thought of as a generalisation of DDIMs, the DDIM-style deterministic sampling effectively implies sampling from a process different from the one with which the model was trained. Thus, for a comparison, we opted to sample from the same processes that the model was trained on, which coincides with the DDPM reverse process. However, since NDMs allow deterministic sampling we acknowledge the interest of this alternative comparison and plan to incorporate it into the camera-ready version of our paper to compare FID scores. The switch from DDPM to DDIM doesn’t affect evaluation of likelihood, since it relies on integration of deterministic process in both cases.
>
>     Nevertheless, in addition to the comparison with DDPM presented in Table 4, we also provide a comparison of NDM with other newer models in Tables 2 and 3.
>
>     Simultaneously, there exist studies proposing techniques to enhance the performance of diffusion models or expedite the sampling procedure [3, 4]. However, most of these approaches are orthogonal to NDMs and can be easily adapted for further gains.
>
> 3. We present the derivation of equation 10 in Appendix A.2. The computation of this equation requires the inference of the transformation $F^\varphi$. Importantly, this does not disrupt the simulation-free paradigm. As described earlier, a non-simulation-free term implies the necessity to simulate certain dynamics during the training procedure. In NDM, we can sample $\mathbf{z}^t$ with just one inference of $F^\varphi$, introducing increased complexity only as a constant without altering the order of complexity.
>
>     In our experiments, the introduction of a learnable $F^\varphi$ extends the training time by approximately 2.3 times. However, in a non-simulation-free approach with $T$ timestamps, employing a learnable forward process would result in a $T$-fold slowdown.
>
>
> [1]:Dongjun Kim et al. 'Maximum Likelihood Training of Implicit Nonlinear Diffusion Models.'
>
> [2]: Tianrong Chan et al. 'Likelihood Training of Schrödinger Bridge using Forward-Backward SDEs Theory.'
>
> [3]: Qinsheng Zhang et al. 'Fast Sampling of Diffusion Models with Exponential Integrator'
>
> [4]: Fan Bao et al. 'Analytic-DPM'

---

> > ### Comment · Reviewer_vgew · 2023-11-20
> > **Questions**
> >
> > Thanks for the responses.
> >
> > 1. I think the performance comparison with DDIM is absolutely necessary in the revision, and I do not see any difficulties in incorporating it in the current revision. I do not understand why the authors hope not to add them until camera-ready version. The checkpoint and code base are public. You can also grab the numerical value from DDIM paper directly.
> > 2. I noticed for Cifar10, the FID for DDPM (11.44) with 1000 steps lags significantly compared with the existing literature (~3.x). Could you explain the reasons?
> > 3. For [1], one needs to propagate the auxiliary network N times in order to get the training source data with size [batch, N, dimension]. I think in your framework, you also need to propagate $F^{\phi}$ N times in order to train the same amount of source data? Though I agree that it is 'simulation-free', it seems not to reduce any training complexity?
> >
> > [1] Tianrong Chan et al. 'likelihood Schrodinger Bridge'

---

> > > ### Author Response · Authors · 2023-11-20
> > >
> > > 1. The switch from DDPM to DDIM only affects the FID score, since the evaluation of the likelihood relies on integration of the deterministic process in both cases. The NLL and NELBO values included in the paper are also representative of DDIM. However, to compare the FID scores of DDIM and NDM, we must compute them for samples drawn from both models using the DDIM-style deterministic sampling regime. Unfortunately, these computations are time-consuming and still running as of this moment.
> > >
> > >     The reason we cannot simply extract numbers from the DDIM paper is that the models are trained with the full ELBO. In contrast, the DDIM paper only provides FID scores for models trained with the simplified $\mathcal{L}_\mathrm{simple}$ objective.
> > >
> > > 2. To ensure that results for NDM and DDPM are comparable, they are both trained with the full ELBO objective (eq. 9). This approach leads to higher FID scores and lower NLL score for conventional diffusion models, which is a well-known attribute of diffusion models trained with the full ELBO.
> > > 3. In the NDM framework, when training a model with $T$ time steps, there is no need to propagate $F^\varphi$ for $T$ times at each step of the training procedure. Instead, the NDM framework enables sampling of the intermediate latent variables $\mathbf{z}^t$ directly from the distribution $q(\mathbf{z}^t|\mathbf{x})$. Therefore, we can maintain the training paradigm outlined in Section 2. Instead of computing all $T$ KL divergences for each time step, we can approximate the objective using the Monte Carlo method by calculating just one KL divergence for a uniformly sampled time step $t \in [1; T]$.
> > >
> > >     This approach allows us to train the model with batches of shape [batch, dimension] rather than [batch, T, dimension]. Consequently, NDM can leverage larger batch sizes and use just one call of $F^\varphi$ for inferring latent variables $\mathbf{z}^t$.

---

> ### Comment · Reviewer_vgew · 2023-11-21
> **Replies**
>
> 1. I totally agree with authors' argument that the full ELBO will lead to lower FID and better NLL, however I still insist that the numerical value FID lags too much. A kind suggestion is that, the authors can demonstrate some experimental setup in which the NLL is more important than other quantitative metrics. The image generation tasks rely quite much on FID because the generated quality counts more. The generated quality is not satisfying even though the higher NLL you achieved (Fig.2).
>
> 2. At the end of the day, the time dimension will be considered as part of batch size for both NDM and SB. The only difference is that NDM chops the time dimension (just randomly sample one timestep which results in [batch, 1,dimension]), and SB chops the batch dimension into small pieces (which results in [small-batch, T, dimension]). I still want to argue that the proposed method is not really 'simulation-free'.
>
> *************EDIT*******************
> My apology for the incorrect term. What I would like to argue is that the complexity will still be high. The proposed method is indeed simulation-free but with comparable complexity like simulation-based approaches (SB).

---

> > ### Author Response · Authors · 2023-11-21
> >
> > 1. In our experiments, the FID scores align with those reported in the literature for conventional diffusion models trained with the full ELBO. Please refer to e.g. [1] (Table 2, Objective $L$, FID is 13.22 for CIFAR-10) or [2] (Table 1 and 2, Objective $\mathcal{L}_\mathrm{vlb}$, FID is 11.47 for CIFAR-10) for specific examples.
> >
> >     Simultaneously, density estimation holds a central role in various applications such as compression [3], semi-supervised learning [4], and adversarial purification [5]. Therefore, we believe that NDM has the potential to enhance performance across a range of practical applications.
> >
> >     We consider NDMs as a comprehensive framework rather than as a specific model that improves the FID score. In contrast to what is often the case in the literature, we do not focus on communicating a single metric such as either FID or the test likelihood but rather consider both. We compare NDMs to conventional diffusions in terms of likelihood estimation, where NDM with learnable transformations consistently demonstrates superior performance across a variety of datasets (including state-of-the-art results for diffusion models on Imagenet 64). Furthermore, we openly acknowledge limitations by reporting FID scores, which are comparable to those of conventional diffusion models under the same conditions. We argue that this a strength of the work and not a weakness.
> >
> >     It is crucial to note that Figure 2 does not showcase the generative performance. In Figure 2, we present qualitative results by demonstrating transformations $F^\varphi(\mathbf{x}, T)$ of data points $\mathbf{x}$ for CIFAR-10 (Figure 2.b) and MNIST (Figure 2.c). Additionally, we provide *one-step* predictions $\hat{\mathbf{x}}^\theta(\mathbf{z}^T, T)$ for CIFAR-10 (Figure 2.d) and MNIST (Figure 2.e). For one-step predictions, the model samples from a standard normal distribution $\mathbf{z}^T \sim \mathcal{N}(0, I)$ and attempts to predict $\mathbf{x}$. Consequently, we do not anticipate these predictions to be of high quality, even though as illustrated by Figure 2.d-e NDM predictions are significantly better. Generative samples for CIFAR-10 are presented in Figure 6 in the Supplementary Material.
> >
> > [1]  Jonathan Ho, Ajay Jain, and Pieter Abbeel. Denoising diffusion probabilistic models. arXiv preprint arXiv:2006.11239, 2020.
> >
> > [2] Alex Nichol and Prafulla Dhariwal. Improved denoising diffusion probabilistic models. arXiv preprint arXiv:2102.09672, 2021.
> >
> > [3] David JC MacKay. Information theory, inference and learning algorithms. Cambridge university press, 2003
> >
> > [4] Z. Dai, Z. Yang, F. Yang, W. W. Cohen, and R. Salakhutdinov. Good semi-supervised learning that requires a bad GAN. In Proceedings of the 31st International Conference on Neural Information Processing Systems, pages 6513–6523, 2017.
> >
> > [5] Y. Song, T. Kim, S. Nowozin, S. Ermon, and N. Kushman. Pixeldefend: Leveraging generative
> > models to understand and defend against adversarial examples. In International Conference on
> > Learning Representations, 2018.

---

> > > ### Author Response · Authors · 2023-11-21
> > >
> > > 2. We apologize for any confusion regarding the terms. Allow us to provide a clearer definition of the term 'simulation-free.’
> > >
> > >     In the literature on diffusion models, the standard definition of the term 'simulation-free' implies that explicit simulation of dynamics is not necessarily required during training to infer an intermediate latent variable  $\mathbf{z}^t$. For instance, in [1], it is necessary to consecutively infer all latent variables  $\mathbf{z}^0, \mathbf{z}^1, \dots, \mathbf{z}^{t-1}$ to sample $\mathbf{z}^t$ making [1] a non-simulation-free approach. In contrast, in NDM, we can directly sample $\mathbf{z}^t$ from $q(\mathbf{z}_t|\mathbf{x})$ without the need to sample any other latent variables.
> > >
> > >     However, we believe that the properties of simulation-free approaches are more crucial than the term itself. Therefore, let us outline the advantages of NDM, specifically as a simulation-free approach.
> > >
> > >     Firstly, we can train the NDM model under the same setup as the SB model [1], with batches of consecutively inferred latent variables of shape [small-batch, T, dimension]. To achieve this, we need to infer latent variables $\mathbf{z}^{0:T}$ according to the definition of the forward process:
> > >
> > >     $$
> > >     q^\varphi(\mathbf{z}^{0:T}|\mathbf{x}) = q^\varphi(\mathbf{z}^T|\mathbf{x}) \prod q^\varphi(\mathbf{z}^{t-1}|\mathbf{z}^t,\mathbf{x})
> > >     $$
> > >
> > >     In this case, we explicitly simulate the dynamics of the forward process. However, this approach comes with several disadvantages. Firstly, neighboring latent variables $\mathbf{z}^{t-1}, \mathbf{z}^t$ are highly correlated and provide almost identical information to the model, rendering them inefficient as a training set. Secondly, we infer all latent variables consecutively here, resulting in $T$ consecutive calls to the neural network in the case of a learnable forward process. This sequential inference makes the process of latent variable inference and the overall training step time-consuming, as observed in SB models like [1].
> > >
> > >     In contrast, NDM, as a simulation-free framework, enables us to switch to sampling latent variables in parallel from the marginal distributions:
> > >
> > >     $$
> > >     \bar{q}^\varphi(\mathbf{z}^{0:T}|\mathbf{x}) = \prod q^\varphi(\mathbf{z}^t|\mathbf{x})
> > >     $$
> > >
> > >     This makes the latent variables $\mathbf{z}^{t-1}, \mathbf{z}^t$ less correlated because we now sample them independently. Additionally, even in the case of a learnable forward process, with this new approach, we can sample all latent variables in parallel with just one call to the neural network.
> > >
> > >     Moreover, thanks to parallel sampling, the simulation-free approach enables us to switch from batches of shape [small-batch, T, dimension] to batches of shape [large-batch, 1, dimension], further diminishing the correlation among latent variables.
> > >
> > >     Therefore, the simulation-free approach significantly reduces the time required for the training step and remarkably decreases the variance of the objective and gradient estimators.
> > >
> > > [1] Tianrong Chan et al. 'Likelihood Training of Schrödinger Bridge using Forward-Backward SDEs Theory.'

---

### Official Review · Reviewer_Y8jp · 2023-10-30

**Soundness:** 3 good
**Presentation:** 2 fair
**Contribution:** 2 fair
**Rating:** 5
**Confidence:** 4

**Summary:**

This paper proposes Neural Diffusion Models, which parameterize the forward diffusion process with a neural network. By training the neural network with the x-prediction network together, the authors yield new generative models that are better than naive diffusion models in terms of likelihood estimation and few-step generation.

**Strengths:**

1. The idea of using parameterized marginal distribution makes a lot of sense, since we have no idea which configuration of the marginal distributions is the best and most of them are handcrafted.

2.The empirical performance successfully demonstrates the effectiveness of the proposed method in likelihood estimation.

**Weaknesses:**

1. The presentation of the paper should be polished. The learning and sampling algorithm is difficult to find. I suggest the authors shorten/defer the discussion section to Appendix and add algorithm boxes in the main text. Moreover, I have several questions on the training process: (1) Are $\phi$ and $\theta$ jointly trained or alternatively trained? (2) How are the hyper-parameters of $F_\phi$ set? Are they similar to the x-prediction network?

2. Certain constraints of $F_\phi(x_t, t)$ should be satisfied. For example, when $t=0$, I think it must satisfy $F_\phi(x_0, 0) = x_0$. I wonder how the authors ensure that.

3. The proposed method brings additional computation overhead in the inference time when simulating Eq. (12) and Eq. (13). Because $F_\phi$ is a neural network and the authors use the same U-Net for both $F_\phi$ and $\hat{x}$ , it at least doubles the computation. Visually, Figure 2 shows that $F_\phi$ actually gives very close prediction to the real data $x$. I guess maybe simply double the size of the original $\hat{x}_\theta$ can get the similar results as NDM.

4. Ablation studies are missing. At least the influence of (1) the various choices of $\alpha_t$ and $\sigma_t$, and (2) the various choices of the network structure and number of parameters of $F_\phi$  should be investigated to verify that NDM brings consistent improvement.

5. Relationship to learnable interpolation in [1] should be discussed in more detail, although I notice there are several sentences in the related works. The two methods are actually very close to each other (I think the only difference is the objective).

Overall, I think using learnable marginal distribution for improve likelihood estimation is reasonable, and there is empirical improvement. However, the poor presentation quality and unstatisfying empirical evaluation makes the paper borderline.

[1] Building normalizing flows with stochastic interpolants. https://arxiv.org/abs/2209.15571

**Questions:**

Please refer to Weakness.

---

> ### Author Response · Authors · 2023-11-20
>
> We thank the reviewer for the suggestions and review which will help to improve the paper. We are glad that the reviewer finds our idea to make sense and empirical results to be illustrative. Below we address the questions and comments raised in the review.
>
> **Weaknesses:**
>
> 1. We agree that adding algorithm boxes can help improve readability. We will include these boxes in the camera-ready version if accepted.
>
>     (1) We optimize the parameters of the forward process $\varphi$ and parameters of the reverse process $\theta$ jointly by optimizing the full ELBO objective (eq. 9).
>
>     (2) As we discuss in Section 4.1, in all our experiments involving learnable transformations, we employ the same neural network architecture and set of hyperparameters for both the transformations $F^\varphi$ and prediction of datapoints $\hat{x}^\theta$.
>
> 2. Upon training NDM with the complete ELBO objective (equation 9), additional restrictions are not necessary. However, as outlined in Appendix C.2, in practice, we ensure that $F^\varphi(\mathbf{x}, t) = \mathbf{x}$ for $t = 0$. This is achieved through the following reparameterization: $F^\varphi(\mathbf{x}, t) = (1 - t) \mathbf{x} + t \bar{F}^\varphi(\mathbf{x}, t)$. This ensures that the reconstruction term $\mathcal{L}^\mathrm{rec}$ doesn’t depend on parameters $\varphi$.
> 3. To address the question of whether the improved performance of NDM is due to the proposed method or merely the result of increasing the number of model parameters we provide an ablation study in Appendix D.3, which demonstrates that the improved quality is not a consequence of an increased number of parameters.
> 4. We introduce NDMs as a framework that generalises existing approaches and enables further extensions through learnable transformations. The main purpose of our experiments is not to obtain state-of-the-art performance nor uniformly better performance compared to baselines. Instead, our aim is to establish the functionality of NDMs in a broader sense, highlighting their additional flexibility and studying learnt transformations.
>
>     We believe that NDM with learnable transformations may exhibit enhanced performance with different neural network architectures, improved parameterization, and after hyperparameter tuning. However, we defer this aspect to future research. In this work, we wanted to keep experimental setup simple to showcase the inherent properties of NDM, prioritizing this over achieving superior results through engineering efforts.
>
>     Nevertheless, as mentioned above, we conducted an ablation study that substantiates the enhanced performance observed in experiments, attributing it to the model itself.

---

> > ### Author Response · Authors · 2023-11-20
> >
> > 5. While NDM and stochastic interpolants [1] are similar in spirit, these methods differ significantly. Firstly, stochastic interpolants represent an approach to learning continuous-time deterministic generative dynamics, whereas NDM learns stochastic dynamics in either discrete or continuous time, which can subsequently may be transformed into a deterministic process.
> >
> >     Secondly, in NDM, the model is trained by optimizing the variational bound on the log likelihood, while stochastic interpolants are trained by optimizing the generalization of the Flow Matching objective [2].
> >
> >     Lastly, NDM joint learns both the forward and reverse processes by optimizing the likelihood, whereas stochastic interpolants learn the generative process with a fixed forward process. The authors of stochastic interpolants demonstrate the possibility of constructing an optimization procedure for the forward process through a max-min game to solve a dynamic optimal transport problem. However, the purpose of this optimization differs from that of NDM.
> >
> >     Moreover, max-min optimization, as employed in stochastic interpolants, is notably less stable compared to min-min optimization in NDM. Additionally, stochastic interpolants do not present experimental results for the optimization of the forward process.
> >
> >     Therefore, we find stochastic interpolants intriguing and promising, as well as related to our work. However, we believe that this approach differs significantly from NDM. We plan to include an extended discussion of stochastic interpolants along with a comparison to NDM in the camera-ready version if accepted.
> >
> > We hope our comments clarified the details of our work. We will keep presentation quality in mind when revising the paper.
> >
> > The primary aim of our experiments section is to illustrate the practical functionality of the NDM framework, complementing the theoretical findings. Additionally, we aim to show that learnable transformations in NDM result in enhanced likelihood estimation while maintaining comparability in terms of FID. We believe that, when combined with improved techniques of parameterization, training, and sampling, NDM may exhibit superior performance. Nevertheless, even with the simplest setup, NDM demonstrates state-of-the-art results on the ImageNet-64 dataset. This is why we present NDM as a framework that introduces flexibility to existing diffusion models rather than as a new model that supersedes others.
> >
> > [1] Building normalizing flows with stochastic interpolants. https://arxiv.org/abs/2209.15571, 2022.
> >
> > [2] Flow matching for generative modeling. arXiv preprint arXiv:2210.02747, 2022.

---

> > > ### Comment · Reviewer_Y8jp · 2023-11-21
> > > **Thank you for the response!**
> > >
> > > I appreciate the additional explanations and clarifications provided by the authors. They correct some of my understandings and  point out experimental evidence in the Appendix.
> > >
> > > My final decision will depend on the overall responses and other questions raised by other reviewers. Thank you once again for the response!

---

### Official Review · Reviewer_LvPx · 2023-10-30

**Soundness:** 3 good
**Presentation:** 3 good
**Contribution:** 2 fair
**Rating:** 6
**Confidence:** 3

**Summary:**

This paper proposes neural diffusion models, or NDMs, extending conventional diffusion models to using learnable non-linear transformation. An objective function is developed within this framework to optimize NDMs, providing an upper bound for the negative log-likelihood.  Empirical studies showcase NDMs' ability to consistently enhance log-likelihood while improving generation quality for scenarios involving a small to medium number of steps.

**Strengths:**

- The background and techniques of NDMs are clearly described.
- The visualization of the transformed data is insightful and pretty helpful in understanding the benefits of NDMs.

**Weaknesses:**

- It seems that the technical details are similar to the conventional DMs. So, what are the technical challenges and novelties here?

- In my opinion, the argument "a key limitation of most existing diffusion models is that they rely on a fixed and pre-specified forward process that is unable to adapt to the specific task or data at hand" is not convincing enough. The extensive empirical studies in the community reflect that conventional DMs have enough flexibility to accommodate diverse data. So, more clearly, I want the authors to clarify what practical consequences would the conventional "inflexible" modeling lead to. On the other hand, conventional modeling bears great simplicity, where the denoising process is training-free. Compared to that, NDMs may rely on more complicated training.

- As mentioned in the paper, the prior term and the reconstruction term should also be trained. What are the weights of these objectives in the entire training objective?

- More clarifications should be put on the training cost and stability of NDMs, compared to vanilla DMs.

- Is there an experiment on a larger dataset using a larger model? Are there technical challenges to achieving this?

**Questions:**

See above

---

> ### Author Response · Authors · 2023-11-20
>
> We are glad that the reviewer founds the description of NDM to be clear and the visualizations to be helpful. Below we address the questions and comments raised in the review.
>
> **Weaknesses:**
>
> - The main novelty of our work is NDM as a simulation-free framework that generalises conventional diffusion models and allows non-linear, time-dependent and learnable transformations of the data distribution in both discrete and continuous time.
>
>     While NDM is a generalisation of conventional diffusion models, it shares many technical similarities with them. However, implementing NDM involves addressing several technical challenges.
>
>     Firstly, in order to consistently define the forward process, we needed to shift to a non-Markovian definition of the model. This required deriving consistent distributions $q(\mathbf{z}_t|\mathbf{x})$ and $q(\mathbf{z}_s|, \mathbf{z}_t, \mathbf{x})$, as well as parameterizing the reverse process and deriving objective.
>
>     Secondly, to extend NDM to continuous time, we derived stochastic and ordinary differential equations (SDE and ODE) along with the corresponding objective. Lastly, for an efficient practical implementation of the continuous-time model, we devised a method to calculate time derivatives $\frac{\partial F_\varphi}{\partial  t}$ using Jacobian Vector Product (JVP). Additionally, we applied importance sampling to ensure the stability of the training procedure.
>
> - Conventional diffusion models indeed exhibit remarkable generalization ability and performance across various domains. However, the forward process in these models implicitly predefines the distribution of latent variables and consequently generative trajectories. The fixed forward process thus determines the complexity of the target for the reverse process and the complexity of the sampling procedure. Consequently, the flexibility of the forward process has the potential to simplify the target for the reverse process, resulting in enhanced performance and simplified trajectories that boost sampling.
>
>     Recent research in diffusion models indicates that techniques such as diffusion in latent or orthogonal space, a combination of diffusion and blurring, flow matching, and rectified flows contribute to improved performance. These techniques bring modifications to the conventional diffusion process, enhancing sample quality and sampling speed. The NDM framework presented in our work represents the next step in the quest for finding better methods to construct generative dynamics.
>
> - We don’t utilize any balancing coefficients. We always train models with the pure ELBO (eq. 9). However, thanks to parameterization of the transformation that guarantees $F_\varphi(\mathbf{x}, t) = \mathbf{x}$ for $t = 0$ we don’t need to optimize the reconstruction term (see Appendix C.2, (eq. 37)).
> - In all our experiments, we employ the same neural network parameterization for both transformations, $F_\varphi$ and prediction of datapoint $\hat{x}_\theta$. Therefore, with this parameterization, NDM has twice as many parameters compared to vanilla diffusion models and approximately 2.3 times longer training duration (refer to Section 6). However, in Appendix D.3, we present an ablation study demonstrating that performance improvements are not achieved by increasing the number of parameters.
>
>     The only additional technique we employed to ensure training stability is importance sampling of time for continuous-time models, as described in Sections 3.2 and 6. No further techniques where necessary to ensure stable training of NDMs. We will clarify it in camera ready.
>
> - In our paper, we present experimental results on the ImageNet 64 and CelebA-HQ-256 datasets, which involve significantly more computational resources compared to experiments on the standard CIFAR-10 dataset. However, across datasets we consistently vary only the scale of the neural network architecture in all our experiments. Thus, the primary technical challenge we identify for experiments on larger datasets with larger models is the associated computational expense.

---

> > ### Comment · Reviewer_LvPx · 2023-11-22
> > **Thanks**
> >
> > Thanks for the answers. I retain my score and will also consider the comments from other reviewers for the final recommendation.

---

### Official Review · Reviewer_rkQA · 2023-10-31

**Soundness:** 2 fair
**Presentation:** 3 good
**Contribution:** 2 fair
**Rating:** 5
**Confidence:** 4

**Summary:**

Diffusion models traditionally use only linear transformations, however using a broader family of transformations can potentially help train generative distributions more efficiently. This work presents Neural diffusion models that generalize existing diffusion models by adding learnable transformation of data which are parameterized with a neural network. The corresponding forward and reverse process are derived by modifying DDPM forward and reverse process. The variational loss objective has been generalized to include learnable transformations. NDM can also be extended to continuous time diffusion models based on previous work by Song et al. 2020. Many previously proposed diffusion models are instances of NDM. Overall, NDM shows gains in NLL and negative ELBO over DDPM.

**Strengths:**

- The idea of generalizing diffusion models to learnable non-linear transformations is interesting.
- Many previously proposed diffusion models and flow models are special cases of NDMs with specific choice of transformation.
- The qualitative results of learned transformations for different datasets in Figure 2 are interesting.
- NDM provides consistent gains in terms of NLL and NELBO over DDPM (See Table 4 and 7).

**Weaknesses:**

- One of the primary motivations for learnable transformations is that it simplifies the data distribution and therefore leads to predictions of x that are more aligned with data. Ideally, if transformations indeed helped with simplification of data distribution, one should have observed better quantitive metrics in fewer sampling steps. However, the actual gains in quantitative metrics like NLL and NELBO seem marginal. Further, there seems to be no consistent gains in terms of FID. In addition, as listed in limitations, the model uses 2.3$\times$ more training time than DDPM (which by itself is slow and needs hundreds of thousands of steps to get good FID). Therefore, I am not sure if the minor gains in NLL and NELBO can be justified when compared to 2.3$\times$ increase in training time as well as $\sim 2\times$ increase in model size.
- Benefits of learnable transformations hasn’t been explored much in the paper and as a result its benefits remain unclear. There is a toy experiment on training optimal transport on 1D data in the appendix. However, larger scale experiments on real data would make this work much stronger. Overall, I feel that NDMs are well-motivated but it remains unclear from the experimental results that introducing learnable transformations as a standard practice for training diffusion models is beneficial.

**Questions:**

- After going through the implementation details (Section 4.1, Appendix C),  the architectural details of neural network used to model the transformation $F_\psi$  remains unclear to me. Could the authors further elaborate on these details?
- Table 4 compares DDPM and NDM. However, NDM has non-Markovian forward and reverse process and for a fair comparison, results for DDIM should also be included, especially in the cases when the number of sampling steps is smaller than the number of training steps.

---

> ### Author Response · Authors · 2023-11-20
>
> We are delighted that the reviewer found our idea of generalization and the qualitative results of learnable transformations intriguing. Below we address the questions and comments raised in the review.
>
> **Weaknesses:**
>
> - As soon as NDM with learnable transformation $F_\varphi$ has a more flexible forward process, an improved likelihood estimation is anticipated, as indeed observed. As illustrated in Section 4.2, the incorporation of learnable transformations consistently enhances both NLL and NELBO across a variety of datasets, including state-of-the-art results on ImageNet-64 dataset.
>
>     While these learnable transformations simplify the generative process, making the approximation by neural networks easier, it does not necessarily ensure the simplicity of generative trajectories. In theory, it may potentially lead to increased curvature of trajectories. Therefore, the results presented for NDM with a reduced number of steps (Table 4, 1000 → 10 steps) are considered more of an empirical finding rather than an expected property.
>
>     While extra flexibility of the forward process leads to consistently improved likelihood estimation, it doesn't always lead to improved FID score. This is a well-known attribute of diffusion models trained with the full ELBO objective (eq. 9).
>
>     We introduce the NDM as a novel framework that generalizes many existing diffusion models and allows for a more complex parameterization of the forward process. Therefore, the primary focus of our experimental section is to demonstrate the feasibility of the framework, rather than presenting a stronger model that surpasses baselines. We believe that the NDM may exhibit enhanced performance with different parameterization and hyperparameter tuning. Therefore, we aimed to maintain a simple experimental setup. Nevertheless, our findings illustrate that the NDM, with learnable transformations, yields improved NLL estimation while remaining comparable in terms of FID score. In addition, in Appendix D.3, we present the ablation study, which demonstrates that the improved quality is not a consequence of an increased number of parameters.
>
> - While we agree that advantages of NDMs are not yet fully exploited, we consider this a strength of the method rather than a weakness. NDM introduces a new flexible framework for diffusion modelling, derives novel time-discrete and time-continuous bounds on the log-marginal likelihood, studies the type of potential transformations that are learned, delivers promising NLL results on a variety of real and synthetic datasets, as well as discusses its potential limitations. Because of the flexibility of NDMs they enable interesting potential for adapting to the problem at hand and new avenues for future research in generative modelling, like learning dynamic optimal transport that we present in Appendix E.
>
> **Questions:**
>
> - In all our experiments with learnable transformations we use the same neural network parameterization for both transformations $F_\varphi$ and prediction of the datapoint $\hat{x}_\theta$.
> - While NDM can be thought of as a generalisation of DDIMs, the DDIM-style deterministic sampling effectively implies sampling from a process different from the one with which the model was trained. Thus, for a comparison, we opted to sample from the same processes that the model was trained on, which coincides with the DDPM reverse process. However, since NDM allows deterministic sampling we acknowledge the interest of this alternative comparison and plan to incorporate it into the camera-ready version of our paper to compare FID scores. The switch from DDPM to DDIM doesn’t affect evaluation of likelihood, since it relies on integration of deterministic process in both cases.

---

> > ### Comment · Reviewer_rkQA · 2023-11-21
> >
> > Dear authors,
> >
> > Thank you for answering my questions! I would like to retain my score for now.

---

### Author Response · Authors · 2023-11-20

We thank the reviewers for their comments and suggestions. We appreciate that the reviewers found the NDM framework to be “interesting”, an “well-motivated”, the method to be “clearly described”, visualisations to be “helpful”, and the effectiveness of the method is “successfully demonstrated”. We believe we have addressed the suggested weaknesses by adding new discussions, clarifications, and suggested revisions to the text.

First, we would like to clarify that we present NDM not as a specific novel model that surpasses all others in terms of FID scores or likelihood estimation. Instead, we introduce NDMs as a comprehensive framework that generalises various existing approaches, enabling the free specification of forward processes and offering additional flexibility by introducing learnable parameters. At the same time, NDMs do not disrupt the standard two processes diffusion paradigm, and we show how they can be generalised to continuous time. This makes them compatible with other approaches that improve sample quality, likelihood estimation or reduce number of diffusion steps.

Nevertheless, our experiments demonstrate that incorporating learnable transformations into the forward process through NDM leads to improved likelihood estimation, including state-of-the-art results on ImageNet-64 dataset. We believe that  better parameterization and hyperparameters tuning could further enhance performance, but we leave this to future work. Moreover, these learnable transformations contribute to better alignment between predictions and the data distribution. Enhanced predictions are valuable, especially in applications such as conditional generation that heavily rely on accurate predictions. NDM also offers a framework for others to integrate the learning of forward noise processes into specific applications, such as protein design, speech synthesis, or medical imaging.

Finally, the additional flexibility opens up the potential for learning novel generative processes with predefined properties, as demonstrated in Appendix E. This is achievable because the forward process can adapt to the constrained reverse process, which is not possible with conventional diffusion.

We have replied to the specific comments and questions of each reviewer individually bellow.

We hope that the new discussion and clarification that the reviews have helped bring about will reinforce or increase your support for acceptance!

---

> ### Author Response · Authors · 2023-11-21
>
> To incorporate the feedback and suggestions from reviewers, we have revised the paper. The key modifications are following:
>
> - In Section 3, we introduced algorithm boxes (Algorithms 1 and 2) outlining the training and sampling procedures within the NDM framework.
> - In Table 4, alongside FID scores for stochastic DDPM-style sampling, we have now included FID scores for deterministic DDIM-style sampling.
> - In Appendix B, we have expanded discussions and comparisons of NDM with other existing models, including Stochastic Interpolants [1] and Schrödinger Bridge-based models.
>
> We thank the reviewers for their valuable assistance in improving the quality of our work.
>
> [1] Building normalizing flows with stochastic interpolants. https://arxiv.org/abs/2209.15571

---

### Author Response · Authors · 2023-11-23

We thank the reviewers for their valuable contributions and discussion, which has enhanced the quality of this work. We summarise the main contributions of NDM and the discussion:

- We present NDMs, a novel general framework for diffusion models, which allows non-linear, time-dependent and learnable transformations of the data distribution.
- We present novel theoretical results, where we derive a simulation-free objective for NDM, which is crucial for efficient training. We also derive ordinary and stochastic differential equations for continuous time sampling. Additionally, we derive discrete time generative dynamics.
- NDM subsumes a large portion of existing approaches as special cases, enabling the free specification of the forward process (Tables 1 and 5). Moreover, it extends these much further, offering additional flexibility by introducing learnable parameters.
- Qualitatively, we explore the properties of NDM with learnable transformations, demonstrating that the introduction of learnable parameters simplifies the distribution and enhances the alignment of predictions with the data (Figure 2).
- NDM, with a simple parameterization of learnable transformations and without hyperparameter tuning, consistently and significantly outperforms baselines in density estimation across a range of small and large-scale experiments (Tables 2-4). Notably, it also achieves new state-of-the-art results on ImageNet-64 and CelebA-HQ-256.
- Our ablation study confirms that the improved quality is not merely a consequence of an increased number of parameters (Table 8).
- NDMs exhibits sample quality comparable to conventional diffusion models under the same conditions. This is evidenced visually (Figure 6) and by FID scores in both stochastic DDPM-style and deterministic DDIM-style sampling (Table 4).
- Other than importance sampling of time, NDMs does not require any additional techniques to ensure stable training.
- We demonstrate the potential of NDM to learn novel generative dynamics, such as dynamic optimal transport (Figure 3), which conventional diffusion models like DDPM and DDIM are incapable of.

Due to NDMs retaining the two process diffusion paradigm it can be readily combined with other approaches such as [1-5] for even further gains.

The extensive range of experiments, including qualitative transformation studies, ablation studies, multiple rigorous image density estimation and generation studies across many different datasets, novel proof-of-concept generative dynamics, show that NDMs is a powerful framework for generative modelling.

NDM allows the user to imbue the generative model with entirely new properties. For example, the ability to learn more straight generative trajectories (Appendix E), or to selectively destroy the information about certain features at different stages of the noising process. The NDM framework opens new ways for extending diffusion models far beyond just being a good generative model.

We trust that the insights gained from our discussion will bolster your support for acceptance!

[1] Jiaming Song, Chenlin Meng, and Stefano Ermon. Denoising diffusion implicit models. arXiv preprint arXiv:2010.02502, 2020a.

[2] Luping Liu, Yi Ren, Zhijie Lin, and Zhou Zhao. Pseudo numerical methods for diffusion models on manifolds. arXiv preprint arXiv:2202.09778, 2022a

[3] Zhisheng Xiao, Karsten Kreis, and Arash Vahdat. Tackling the generative learning trilemma with denoising diffusion GANs. arXiv preprint arXiv:2112.07804, 2021.

[4] Tim Salimans and Jonathan Ho. Progressive distillation for fast sampling of diffusion models. arXiv preprint arXiv:2202.00512, 2022.

[5] Kim, Dongjun, et al. "Consistency Trajectory Models: Learning Probability Flow ODE Trajectory of Diffusion." *arXiv preprint arXiv:2310.02279* (2023).

---

### Meta-Review · Area_Chair_neSC · 2023-12-09

**Metareview:**

The paper explores the possibility and benefits of allowing the forward process of a diffusion model to also be learned. The development is based on adding a parameterized nonlinear transformation on the conditioned data variable in the DDIM formulation. The corresponding training and sampling methods and the continuous-time counterpart are derived, in which the "simulation-free" desideratum for training is kept. The method achieves improved likelihood on image datasets, and shows improved generation quality under the constraint of fewer diffusion steps. The reviewers acknowledged this exploration to extend diffusion models, and the presented benefits. It is informative that authors showed that improvement is not just from more parameters.

Nevertheless, the reviewers still seem not fully convinced of the value of the proposed method. Although conventional diffusion models use a prespecified forward process which seems to be less flexible, the process provides a clear guidance that breaks down the complicated end-to-end distribution transformation into a set of much simpler transformations. This is an assumed reason that diffusion models outperform e.g. VAE, GAN, flow models. The reviewers found the empirical performance is not consistently better than conventional diffusion models, so the concern is not completely answered. Reviewers also raised concerns for a more clear and proper discussion with related works.

**Justification For Why Not Higher Score:**

After authors had responded to the raised concerns, all reviewers replied, which indicates that the concerns still remain, and the weaknesses is outweighed over the strengths.

**Justification For Why Not Lower Score:**

N/A

---

### Decision · Program_Chairs · 2024-01-16

Reject